# Exact Unlearning in Reinforcement Learning

**Thanh Nguyen-Tang** [1]   **Raman Arora** [2]

## Abstract

We formulate the problem of *exact unlearning* in reinforcement learning, where the goal is to design an efficient framework that enables the removal of any user's data upon deletion request, i.e., the online learner's output after unlearning is *indistinguishable* from what would have been produced had the deleted user never interacted with the learner. For any $\rho > 0$, we show that there exists a reinforcement learning (RL) algorithm that is $\rho$-TV-stable and supports an exact unlearning procedure whose expected computational cost is only a $\rho\sqrt{\ln T}$ fraction of the computational cost of retraining from scratch. We construct such a $\rho$-TV-stable RL algorithm for tabular Markov decision processes (MDPs), which achieves a regret bound of $\mathcal{O}(H^2\sqrt{SAT} + H^3S^2A + H^{2.5}S^2A/\rho)$, where $S$, $A$, $H$, and $T$ denote the number of states, the number of actions, the episode horizon, and the number of episodes, respectively. We also establish a lower bound of $\Omega(H\sqrt{SAT} + SAH/\rho)$ for $\rho$-TV-stable RL algorithms, showing that our algorithm is nearly minimax optimal.

## 1. Introduction

Machine unlearning is a relatively new area of research that aims to efficiently remove the influence of specific data from a machine learning (ML) model in response to a data modification or deletion request (Cao & Yang, 2015; Bourtoule et al., 2021). This need arises from growing privacy concerns and legal requirements mandating that certain user data, along with its effect on a model, be completely removed. Such concerns are motivated by the vulnerability of ML models to attacks such as membership inference (Shokri et al., 2017) and model inversion (Fredrikson et al., 2015),

which can leak sensitive training data.

Moreover, various data protection laws have enshrined users' *right to be forgotten*, including the General Data Protection Regulation (GDPR) in the European Union (GDPR, 2018), the California Consumer Privacy Act (CCPA) (Bonta, 2022), the Act on the Protection of Personal Information (APPI) in Japan (JDPO, 2019), and Canada's proposed Consumer Privacy Protection Act (CPPA) (CPPA, 2023). Beyond regulatory compliance, machine unlearning also offers practical benefits: it can be used to remove poisoned clients or compromised nodes in federated systems (Jin et al., 2023), delete copyrighted or proprietary content from models (Eldan & Russinovich, 2023), accelerate leave-one-out validation, support user data marketplaces, and identify high-value data points within a model (Ginart et al., 2019, p. 2).

Machine unlearning requires that the model's output after unlearning be *indistinguishable* from the output that would have been produced had the requested user data never been included in the training process. While there is no universally accepted definition of indistinguishability in this context, two primary notions of certified unlearning have emerged: exact unlearning (Ullah et al., 2021; Ullah & Arora, 2023) and approximate unlearning (Guo et al., 2019; Neel et al., 2021; Sekhari et al., 2021; Allouah et al., 2024; Van Waerebeke et al., 2025).

**Why exact unlearning in RL?**   Approximate unlearning is less stringent and typically enables more efficient algorithms with improved space complexity. However, it does not guarantee full removal of a user's influence, which can be problematic in practice, particularly because it is difficult to determine an appropriate approximation parameter that ensures adequate privacy protection. As a result, organizations that train large-scale models may prefer exact unlearning, accepting the cost of increased memory usage during training in order to avoid potentially catastrophic consequences of a privacy breach, especially if even a single individual can demonstrate that their personal data remains embedded in a deployed model. This is particularly relevant for interactive systems (recommendation, personal assistants, healthcare triage) that continually log per-user episodes. Deletion requests arise for privacy (right-to-be-forgotten, membership inference risk), safety (removing poisoned/outlier interactions), compliance (per-user auditabil-

[1]Department of Data Science, New Jersey Institute of Technology, Newark, NJ, USA [2]Department of Computer Science, Johns Hopkins University, Baltimore, MD, USA. Correspondence to: Thanh Nguyen-Tang <thanh.nguyen@njit.edu>, Raman Arora <arora@cs.jhu.edu>.

*Proceedings of the 43rd International Conference on Machine Learning*, Seoul, South Korea. PMLR 306, 2026. Copyright 2026 by the author(s).

ity), and engineering (fast leave-one-user-out diagnostics) reasons. Our framework lets operators retrofit a strong deletion guarantee on to regret-optimal learners while keeping retraining rare and localized, which is valuable even in tabular settings that back many real-world pipelines (e.g., bandits with context bucketing).

To date, most work on machine unlearning has focused on supervised and unsupervised learning settings, where data points are static and independently processed. However, many real-world systems, such as recommender platforms, digital assistants, and personalized healthcare tools, are interactive in nature and rely on reinforcement learning (RL) to model sequential user interactions. In these systems, data arises not from isolated inputs but from temporally extended experiences with users.

RL is a fundamental paradigm for sequential decision-making, where an agent learns to maximize cumulative reward in an unknown environment through trial and error. With its growing adoption in personalized services, ranging from online recommendations to virtual assistants and social robotics, RL algorithms increasingly interact with streams of users, adapting continuously based on their behaviors. This naturally raises the same privacy concerns central to machine unlearning: *how can we remove the influence of a particular user's interaction history from an RL system when requested?*

Our motivation stems from personalized, interactive systems, e.g., voice assistants, recommender platforms, medical data trajectories, or personalized tutors, where episodic RL interactions correspond to identifiable users. Here is a concrete example inspired by the work of (Shani et al., 2005).

*Example* 1.1. Recommender systems (e.g., product recommendations on e-commerce platforms such as Amazon) are often modeled as MDPs, where an RL agent acts as the recommender. The system interacts sequentially with different users, each corresponding to an episode $t$. An action is an item recommendation, and the user's state at episode $t$ and step $h$ can be represented by their last $k$ selected items, i.e., $s_h = (a_{h-k}, \ldots, a_{h-1})$, since recent history is most relevant for prediction. Given a recommendation $a$, the user may accept it, transitioning from $s_h = (a_{h-k}, \ldots, a_{h-1})$ to $s_{h+1} = (a_{h-k+1}, \ldots, a_{h-1}, a)$, or instead select a non-recommended item $a'$, transitioning to $s_{h+1} = (a_{h-k+1}, \ldots, a_{h-1}, a')$. The reward reflects the utility of selling items to the user, while the *sampled* rewards and transitions at episode $t$ capture the characteristics of the user interacting with the system. A user who interacts with the recommender at time $t$ may later request removal of her interaction data, due to privacy or other concerns. Crucially, it is not enough to delete the raw data, we must remove its influence on the trained system.

Despite the relevance and urgency of this question, the prob-

lem of unlearning in RL remains largely unaddressed. In this work, we aim to bridge this gap by formulating and addressing the problem of exact unlearning in RL (see Definition 2.1). Our goal is to design sample-efficient RL algorithms that enable the efficient removal of any user's data upon request, while ensuring that the resulting model behavior is indistinguishable from one trained without that data. Our key results are as follows:

1. We formulate the problem of exact unlearning in reinforcement learning. To do so, we abstract RL into a general class of sequential learning problems with prefix sum structure, and develop a unified (un)learning framework based on the recent notion of Total Variation (TV) stability (Ullah et al., 2021; Ullah & Arora, 2023) (see Section 3).

2. We show that, for any $\rho > 0$, there exists an efficient RL algorithm for tabular Markov decision processes (MDPs) that is $\rho$-TV-stable (see Definition 2.2) and achieves a regret bound of $\tilde{\mathcal{O}}\left(H^2\sqrt{SAT} + H^3 S^2 A + \frac{H^{2.5}S^2 A}{\rho}\right)$, where $S, A, H, T$ denote the number of states, the number of actions, the horizon, and the number of episodes, respectively (see Section 4). This algorithm admits an efficient exact unlearning algorithm whose expected computational cost is only $\rho\sqrt{\ln T}$ fraction of the cost of retraining from scratch (see Section 3).

3. We derive a minimax lower bound of $\Omega\left(H\sqrt{SAT} + \frac{HSA}{\rho}\right)$ for the class of $\rho$-TV-stable RL algorithms, demonstrating that our upper bound is nearly tight (see Section 4.1).

**Overview of Techniques.** Our work builds on Ullah & Arora (2023) who showed that TV-stability is both sufficient and necessary for exact unlearning in batch supervised learning in a natural coupling-based learning framework. Extending this theory to sequential RL is highly non-trivial as it requires a) establishing TV-stable RL algorithms and (b) proving that these algorithms simultaneously achieve near-optimal regret.

We extend UCB-VI (Azar et al., 2017) in a non-trivial manner to achieve exact unlearning. Our RL algorithm is the first regret-optimal variant that is also $\rho$-TV-stable and supports exact unlearning. This requires (i) replacing visitation statistics with binary-tree, noise-perturbed prefix sums while preserving optimism, (ii) storing intermediate sufficient statistics in a coupling-compatible way so that unlearning can reuse randomness via maximal coupling, (iii) extending regret proof to control additional variance from correlated Gaussian noise, matching UCB-VI up to log factors.

We also establish the first minimax lower bound for the regret of TV-stable algorithms. Our contribution thus lies in

integrating and extending these foundational ideas into the RL setting with explicit algorithms and rigorous theoretical analysis, filling a significant gap in the literature.

**Related Work.** In the broader landscape, a number of important developments have emerged across different learning tasks. While we do not attempt to be exhaustive, we highlight several key contributions. The term machine unlearning was first introduced by Cao & Yang (2015), who proposed a deterministic notion of data deletion in trained models. Their work focused on statistical query problems under restrictive structural assumptions. Ginart et al. (2019) initiated the study of approximate unlearning via differential privacy, focusing on the $k$-means problem. This line of work was extended to linear and logistic regression by Guo et al. (2019), and to general convex models by Neel et al. (2021).

The work most closely related to ours is that of Ullah et al. (2021); Ullah & Arora (2023). These works formalize the notion of exact unlearning in the batch setting, particularly in connection with adaptive query release mechanisms. In contrast, we study exact unlearning in the context of sequential learning, specifically within reinforcement learning and regret minimization. While our setting is distinct, we adopt their notion of exact unlearning and build on similar algorithmic primitives, most notably, the use of prefix sums, to support efficient unlearning in an online framework.

In the context of RL, prior work on unlearning is extremely limited. To our knowledge, the only relevant effort is by Ye et al. (2023), who introduced the concept of reinforcement unlearning. However, in their setting, multiple distinct MDPs are trained simultaneously (e.g., for different tasks or domains), and unlearning involves removing an entire MDP rather than individual user episodes within a single MDP. In contrast, we consider a more granular and practical objective: unlearning at the level of an individual user's interaction. Furthermore, while Ye et al. (2023) empirically study approximate unlearning, they do not provide theoretical guarantees or analyze its effect on regret. Our work differs in that we provide a finite-sample regret bound for exact unlearning, along with a matching lower bound.

Finally, our work is related to the growing literature on differentially private reinforcement learning (Vietri et al., 2020; Zhou, 2022; Chowdhury & Zhou, 2022; Qiao & Wang, 2023). While differential privacy shares conceptual similarities with approximate unlearning (indeed, some techniques overlap), its guarantees are fundamentally different. Differential privacy requires that model outputs be statistically similar whether or not a single datapoint is included in the training set. In contrast, exact unlearning demands that the resulting model be identically distributed to one trained without the deleted data. As such, results in differential privacy and approximate unlearning do not directly imply guarantees for exact unlearning. Nevertheless, techniques developed in the differential privacy literature, such as the binary tree mechanism, play a crucial role in our framework.

While preparing this paper, we became aware of parallel and independent work by Hu et al. (2025), which studies unlearning in the setting of online convex optimization. While both works address unlearning in online learning, there are two key distinctions. First, our work focuses on *exact* unlearning, whereas Hu et al. (2025) develop methods for *approximate* unlearning. Second, we analyze unlearning in the context of MDPs, a structured, sequential setting that introduces additional challenges, while their work is situated in the stateless framework of online convex optimization.

## 2. Problem Setup and Preliminaries

Sequential learning problems arise in settings where an online stream of users (or clients) arrives over time, and the model must make real-time, personalized decisions based on accumulated experience. Common applications include online recommendation systems, virtual assistants, and adaptive educational platforms, where each user interaction provides feedback that informs future predictions or recommendations. Formally, let $\mathcal{Z}$ denote the user space, where each user $z_t \in \mathcal{Z}$ represents an individual arriving at time $t$. The learner maintains a model $w_t \in \mathcal{W}$ drawn from a model space $\mathcal{W}$, which is used to interact with user $z_t$. In addition to the model, the learner may maintain auxiliary meta-data $m_t \in \mathcal{M}$ where $\mathcal{M}$ denotes the meta-data space (e.g., state visit counts, confidence intervals, or cumulative statistics).

Upon interacting with the model, each user produces a response $x_t \in \mathcal{X} \subseteq \mathbb{R}^d$, which captures structured feedback from the interaction, such as observed rewards, state transitions, or implicit preferences. This response is modeled by an **environment oracle** $\{\mathbb{EO}_t\}_{t \geq 1}$, which maps the user and the current model history to a feedback statistic:

$$x_t = \mathbb{EO}_t(z_t; w_{1:t}),$$

where $w_{1:t} = \{w_1, w_2, \ldots, w_t\}$ denotes the sequence of models deployed thus far.

For example, in an online movie recommendation system, each user $z_t$ could represent an individual viewer with latent preferences. The model $w_t$ might encode a recommendation policy or ranking of items personalized to the current user base. Upon presenting recommendations to $z_t$, the system observes their clicks or ratings, which are summarized in $x_t$, such as which movies were selected, how long they were watched, or how they were rated. This user feedback, formalized as $x_t = \mathbb{EO}_t(z_t; w_{1:t})$, is then used to update the learner's model going forward. The overarching goal of the online learner $\mathbf{A} : \mathcal{Z}^* \to \mathcal{W}^* \times \mathcal{M}^*$ is to produce a sequence of models $\{w_1, \ldots, w_t, \ldots\} \subset \mathcal{W}$ that improve

over time, continually adapting to better serve the evolving user population.

**Markov Decision Processes.** Formally, we study sequential learning problems in the framework of reinforcement learning, wherein the user behavior is modeled using a Markov decision process (MDP). Specifically, an MDP is a tuple $M = (\mathcal{S}, \mathcal{A}, H, \{\mathbb{P}_h\}_{h\in[H]}, \{\bar{r}_h\}_{h\in[H]})$ where $\mathcal{S}$ is the finite state space with $S = |\mathcal{S}|$, $\mathcal{A}$ is the finite action space with $A = |\mathcal{A}|$, $H$ is the episode horizon, $\bar{r}_h : \mathcal{S} \times \mathcal{A} \to [0,1]$ is the expected reward function at step $h$, and $\mathbb{P}_h : \mathcal{S} \times \mathcal{A} \to \Delta(\mathcal{S})$, defines the transition dynamics at step $h$, where $\Delta(\mathcal{S})$ denotes the set of all distributions over $\mathcal{S}$.

**Interaction Protocol.** At each episode $t$, the RL algorithm interacts with a *randomly drawn* user (data) $z_t := \{(s_h^t, r_h^t)\}_{h\in[H]} \in \mathcal{Z}$ and recommends a sequence of actions $a_1^t, \ldots, a_H^t$. We assume that the user data is randomly drawn according to the underlying MDP transition kernel and reward function: $s_{h+1}^t \sim \mathbb{P}_h(\cdot|s_h^t, a_h^t)$, $r_h^t \sim R(\bar{r}_h(s_h^t, a_h^t))$, and $R(r)$ is a reward distribution over $[0,1]$ with mean $r$.[1] This user-episode abstraction is commonly adopted in the literature on privacy-preserving RL (Vietri et al., 2020; Zhou, 2022; Chowdhury & Zhou, 2022; Qiao & Wang, 2023) and MDP-based personalized recommendation systems (Shani et al., 2005), where users generate distinct and identifiable reward/transition feedback over time (also see Example 1.1). Crucially, there is a single fixed tabular MDP $(\mathcal{S}, \mathcal{A}, H, P_h, r_h)$ throughout. A "user" $z_t$ is purely a label for episode $t$ that facilitates deletion requests; it is not a latent parameter that changes the environment across episodes. Each episode generates a trajectory under the current policy in the same MDP. The environment oracle $\mathbb{EO}_t$ simply packages that episode's trajectory statistics $x_t$ for our prefix-sum machinery.

**Policies and Value Functions.** A (non-stationary) policy $\pi = (\pi_1, \ldots, \pi_H)$ consists of decision rules $\pi_h : \mathcal{S} \to \Delta(\mathcal{A})$, where $\pi_h(a|s)$ defines the distribution over actions at step $h$ in state $s$. The value function of policy $\pi$ at step $h$ is defined as $V_h^\pi(s) := \mathbb{E}_\pi[\sum_{h'=h}^H r_{h'}|s_h = s]$ where the expectation is over the randomness of the trajectory induced by following $\pi$. The corresponding action-value function is $Q_h^\pi(s,a) = \mathbb{E}_\pi[\sum_{h'=h}^H r_{h'}|(s_h, a_h) = (s,a)]$. We denote by $\pi^*$ an optimal policy and write $V^* = V^{\pi^*}$ and $Q^* = Q^{\pi^*}$. An RL algorithm $\mathbf{A}$ is an *instantiation* of the online learner $\mathbf{A} : \mathcal{Z}^* \to \mathcal{W}^* \times \mathcal{M}^*$ we defined earlier in this section. In particular, for this instantiation of the online learner to MDPs, *the model space $\mathcal{W}$ becomes the*

---

[1] We view a user data $z \in \mathcal{Z}$ (at an episode $t$) as a depth-$H$ tree, which defines the sequence of all possible reward-state pairs $(r_h, s_{h+1})_{h\in[H]}$ in response to any RL algorithm's action sequence $(a_1, \ldots, a_H) \in \mathcal{A}^H$.

*set of all policies* and *the meta-data space $\mathcal{M}$ captures any auxiliary information that is produced by the RL algorithm* (e.g., visitation counts, noise terms, sufficient statistics).

**Regret Minimization.** The goal of the RL algorithm is to minimize the regret over $T$ episodes:

$$R(T) = \sum_{t=1}^T (V_1^*(s_1^t) - V_1^{\pi^t}(s_1^t))$$

where $\pi^t$ is the policy used in episode $t$, and $s_1^t$ is the initial state of that episode.

In a continual stream of user interactions, it is natural to consider scenarios where a user requests the removal of their data from the system. This request requires not only the deletion of the user's data but also the removal of its influence on the trained model. An unlearning algorithm is designed to facilitate this process, ensuring that the model behaves as if the deleted data had never been seen.

Formally, we define an *unlearning algorithm* $\mathbf{U} : \mathcal{W}^* \times \mathcal{M}^* \times \mathcal{Z} \times \mathcal{Z} \to \mathcal{W}^* \times \mathcal{M}^*$, which takes as input a sequence of models and meta-data, along with a target user $z \in \mathcal{Z}$ to be deleted and a dummy user $z' \in \mathcal{Z}$ used as a replacement. The algorithm then outputs an updated sequence of models and meta-data in which the influence of $z$ has been effectively removed and replaced by the influence of $z'$.

To formalize this goal, we introduce a notion of *exact unlearning* for sequential learning problems, which we directly adopt from the batch learning setting developed by Ullah & Arora (2023). This definition captures the requirement that the output of the unlearning algorithm must be *indistinguishable in distribution* from the output that would have resulted had $z$ never been observed.

*Definition* 2.1 (**Exact Unlearning for Sequential Learning Problems**). An unlearning algorithm $\mathbf{U}$ is said to achieve *exact unlearning* with respect to a learning algorithm $\mathbf{A}$ if, for any $T \in \mathbb{N}$, any $t \in [T]$, and any sequence $z_1, \ldots, z_T, z'$ in $\mathcal{Z}$, the following holds

$$\mathbf{U}(\mathbf{A}(z_{1:T}), z_t, z') \overset{D}{=} \mathbf{A}(z_1, \ldots, z_{t-1}, z', z_{t+1}, \ldots, z_T),$$

where $X \overset{D}{=} Y$ denotes equality in distribution, meaning that for any event $\mathcal{E}$, $\Pr(X \in \mathcal{E}) = \Pr(Y \in \mathcal{E})$.

Definition 2.1 formalizes the idea that deleting a user data point $z_t$ is equivalent to having replaced it with an arbitrary but fixed data point $z'$ from the outset. The requirement that the equivalence must hold for any choice of $z'$ ensures that the notion of deletion is meaningful and practically aligned with privacy expectations. The dummy data point $z'$ can be chosen by the algorithm designer and need not correspond to a real user; for instance, it may encode a no-op or neutral interaction.

Replacing data rather than removing it outright has a key algorithmic advantage. It allows the learner to preserve any internal data structures (e.g., prefix sums, counters, or trees) without disrupting their structure. This structural preservation is crucial for enabling efficient unlearning in online settings.

*Remark* 2.1. In batch learning settings, the replacement data point is typically sampled randomly from the dataset (Ullah & Arora, 2023). In contrast, in the sequential setting, we explicitly replace a deleted data point with a fixed dummy data point chosen by the algorithm designer.

**Unlearning Request.** We consider the *anytime* deletion request setting, in which a user may request deletion at any time $T$, without the learner knowing $T$ in advance. While $T$ is independent of the learner's algorithm, it necessitates the design of both anytime learning and unlearning algorithms.

*Remark* 2.2 (Anytime vs. fixed-time). Although we technically operate in the anytime setting, for clarity and without loss of generality, we may assume a fixed-time deletion request, where $T$ is known. This simplification is justified because the binary tree mechanism of (Dwork et al., 2010), which we employ, does not require the tree's size to be known in advance. Furthermore, standard techniques such as the doubling trick (Auer et al., 1995) allows us to convert regret bounds with known time horizon to anytime regret bounds at the cost of a negligible constant.

A trivial unlearning approach is to retrain the model from scratch on the modified sequence $z_1, \ldots, z_{t-1}, z'$, $z_{t+1}, \ldots, z_T$. While this satisfies the exact unlearning in Definition 2.1, its computational cost is often prohibitive in practice. In many real-world applications, e.g., large-scale recommendation systems or continual learning environments, retraining from scratch for each deletion request is simply infeasible. Our goal is to develop more efficient unlearning methods that are significantly less costly than full retraining.

To motivate such efficient algorithms, we turn to the concept of **algorithmic stability**, specifically in terms of Total Variation (TV) distance, a technique recently leveraged for certified unlearning in the batch setting (Ullah et al., 2021; Ullah & Arora, 2023). The Total Variation (TV) distance between two probability distributions $P$ and $Q$ is defined as

$$\mathrm{TV}(P, Q) := \sup_{\text{measurable event } \mathcal{E}} |P(\mathcal{E}) - Q(\mathcal{E})|.$$

*Definition* 2.2 ($\rho$-TV-stable algorithms). Fix any $\rho \geq 0$. An online learning algorithm $\mathbf{A}$ is said to be $\rho$-TV-stable if, for any two user sequences $Z$ and $Z'$ of the same length that differ in exactly one position, we have: $\mathrm{TV}(\mathbf{A}(Z), \mathbf{A}(Z')) \leq \rho$. Unless noted otherwise, all probabilities are over the algorithm's internal randomness; events are Borel sets in the output space of ($w_{1:T}$, meta-data).

---

**Algorithm 1** Online algorithms with prefix sums

**Require:** environment oracle $\{\mathbb{EO}_t\}$ and the update function **UPDATE** internal to the algorithm
1: Initialize $w_1 \in \mathcal{W}$ and $u_0 = 0$
2: **for** $t = 1, 2, \ldots,$ **do**
3:     A user $z_t$ (oblivious to the algorithm) arrives
4:     Use $w_{1:t}$ to interact with user $z_t$ via the environment oracle $x_t \leftarrow \mathbb{EO}_t(z_t; w_{1:t})$
5:     Update the prefix sum $u_t \leftarrow u_{t-1} + x_t$
6:     Update model via the update function $w_{t+1} = $ **UPDATE**$(u_t; w_{1:t})$
7: **end for**
**Ensure:** $w_0, w_1, \ldots$

---

**Maximal Coupling for Efficient Unlearning via TV-stability.** Let $P, Q$ denote the distributions over the outputs of a learning algorithm run on the original data sequence and the sequence after a deletion request, respectively. Suppose we have access to a sample $x \sim P$. The goal of unlearning is to transform this sample into one from $Q$, using an edit function $\phi : \mathrm{dom}(P) \to \mathrm{dom}(Q)$ such that $y := \phi(x) \sim Q$, and such that the computational cost of applying $\phi$ is small. Ullah et al. (2021) propose a general framework based on *maximal coupling* for constructing such efficient unlearning mechanisms. A coupling of $P$ and $Q$ is a joint distribution $\psi$ with marginals $P$ and $Q$. A *maximal coupling* is a coupling that minimizes the probability of disagreement between the coupled random variables, i.e., $\Pr_{(x,y)\sim\psi} \mathbb{1}\{x \neq y\} = \mathrm{TV}(P, Q)$. This implies that, if the edit function $\phi$ induces a maximal coupling, then with probability at least $1 - \mathrm{TV}(P, Q)$, the original sample $x \sim P$ can be reused directly as a valid sample for $Q$, avoiding retraining.

The key algorithmic challenge is twofold: (1) design learning algorithms that are both accurate and exhibit small TV-stability, and (2) construct edit functions that induce or approximate maximal couplings. These components together enable fast and exact unlearning in the sequential setting.

## 3. (Un)Learning for Prefix Sums

In this section, we consider a class of online algorithms in Section 2, but with an additional structure of **prefix sums**. This algorithm class captures most of online learning algorithms, including reinforcement learning that we will consider later. We present a generic template for online algorithms with prefix sums in Algorithm 1. A key structure of an algorithm in this class is an internal update function **UPDATE** in Line 6. The internal update takes in the prefix sum $u_t$ of the statistics queried from the previous users $z_{1:t}$ and the output of the previous models $w_{1:t}$ to produce a new model output $w_{t+1}$.

## 3.1. TV-stable Learning with Binary Tree Mechanism

We now present Algorithm 2, a modification of Algorithm 1 that makes the algorithm TV-stable and supports efficient unlearning in what follows. The lines 4 and 5 in Algorithm 2 are the only major modifications on top of Algorithm 1.

The idea is that we use the standard binary tree mechanism in the differential privacy literature (Dwork et al., 2010) to add noise to the prefix sums, as has been done in the original exact unlearning framework of (Ullah & Arora, 2023) (see examples in (Ullah & Arora, 2023)). Intuitively, to perturb $T$ prefix sums $\{u_t\}_{t \in [T]}$, one need not use $T$ independent noise samples for each statistic $x_t$. Instead, we only need to add noises to all the dyadics and reuse these noises for any prefix sum based on its dyadic decomposition. For example, suppose we want to add noise to the seventh prefix sum $u_7$, we use the dyadic decomposition of 7 as $7 = 4 + 2 + 1$, and obtain a perturbed variant of $u_7$ by $(x_1 + x_2 + x_3 + x_4 + \xi_1) + (x_5 + x_6 + \xi_2) + (x_7 + \xi_3)$ where $\xi_1, \xi_2, \xi_3 \sim \mathcal{N}(0, \sigma^2 I_d)$. Overall, each prefix sum is perturbed by at most $\log T$ noise samples. This idea can be realized by considering a perfect binary tree $\mathcal{T}$ with the following properties. Each node $b$ of the tree is associated with a fixed, independent sample of Gaussian noise $\xi_b \sim \mathcal{N}(0, \sigma^2 I_d)$. We store in each node $b$ two fields: clean value $v_b \in \mathbb{R}^d$ and noisy value $\tilde{v}_b \in \mathbb{R}^d$, where $\tilde{v}_b = v_b + \xi_b$. A leaf node $t$ additionally stores model $w_t$. The clean value of a leaf node $t$ is zero if in initialization, or $x_t$, if the leaf node gets an update from $x_t \leftarrow \mathbb{EO}_t(z_t; w_{1:t})$. For a non-leaf node, its clean value is the sum of the clean values of its children. The tree supports the following operations:

- **TreeUpdate**$(x; t, \mathcal{T})$: If leaf $t$ does not exist, create leaves $t, t+1, \ldots, 2^{\lceil \log_2 t \rceil}$. Iterate over each node through the path from leaf $t$ to root, and update its clean value by adding the new value and subtracting the old value of leaf $t$. Consequently, the new clean value of leaf $t$ is $x$ and the value of any other leaves in $t+1, \ldots, 2^{\lceil \log_2 t \rceil}$ is zero.
- **GetNoisyPrefixSum**$(t, \mathcal{T})$: Retrieve the noisy version of the prefix sum $\sum_{i=1}^{t} x_i$ from the tree. To do this, write $t = b_1 \ldots b_K$ as $K$-bit representation, where $K$ is the depth of the tree. Initialize $u = 0$. Iterate through the path from the root to leaf $t$ (i.e., for $k = 1, \ldots, K$). If $b_k = 0$, skip to $k+1$. If $b_k = 1$ and the current node is a left child of a node, add to $u$ the current node's noisy value. If $b_k = 1$ and the current node is a right child of a node, add to $u$ the current node's left sibling's noisy value.

## 3.2. Unlearning by Maximally Coupling a Binary Tree

We now present an unlearning framework in Algorithm 3 for Algorithm 2. The idea largely follows from the original unlearning framework of (Ullah & Arora, 2023), by max-

imally coupling the binary tree returned by Algorithm 2. In fact, we can view the unlearning framework here as a simplified version of the unlearning framework of (Ullah & Arora, 2023), whereas due to the sequential nature of our

---

**Algorithm 2** `TV-stable-Learn`$(t_0, T)$

**Require:** binary tree $\mathcal{T}$, initial episode $t_0$, last episode $T$, noise variance $\sigma$.
1: If the binary tree has more than $2^{\lceil \log_2 t_0 \rceil} + 1$ leaves, truncate it so that the tree has only $2^{\lceil \log_2 t_0 \rceil}$ leaves. If the leaves $t_0, t_0 + 1, \ldots$ exist in the tree $\mathcal{T}$, then iterate over all the leaves from $t_0$ onward, set the value of each of the leaves to zeros, and update all the non-leaf nodes
2: **for** $t = t_0, \ldots, T$ **do**
3:     $x_t \leftarrow \mathbb{EO}_t(z_t; w_{1:t})$
4:     **TreeUpdate**$(x_t, t; \mathcal{T})$
5:     $\tilde{u}_t \leftarrow$ **GetNoisyPrefixSum**$(t, \mathcal{T})$
6:     $w_{t+1} \leftarrow$ **UPDATE**$(w_{1:t}, \tilde{u}_t)$
7:     Save $w_t$ to leaf $t$
8: **end for**
**Ensure:** $\mathcal{T}$

---

**Algorithm 3** `Unlearn`$(\mathcal{T}, t)$

**Require:** a binary tree $\mathcal{T}$, the position $t$ to be unlearned, a dummy user $z'$
1: Retrieve $x_t$ from leaf $t$
2: Engage the environment oracle at time step $t$: $x'_t = \mathbb{EO}_t(z'; w_{1:t})$
3: Set $b = t$
4: **while** $b \neq \emptyset$ **do**
5:     Retrieve $(v_b, \tilde{v}_b)$ from $b$ and compute $v'_b \leftarrow v_b - x_t + x'_t$
6:     **if** `Unif`$(0,1) \leq \frac{f_{\mathcal{N}(v'_b, \sigma^2 I)}(\tilde{v}_b)}{f_{\mathcal{N}(v_b, \sigma^2 I)}(\tilde{v}_b)}$ **then**
7:         $v_b \leftarrow v'_b$
8:     **else**
9:         $\tilde{v}'_b \leftarrow v'_b + v_b - \tilde{v}_b$
10:         $(v_b, \tilde{v}_b) \leftarrow (v'_b, \tilde{v}'_b)$
11:         Call **TV-stable-Learn**$(t', T)$ (Algorithm 2), where $t'$ is the leaf that is right after node $b$ in the post-traversal order.
12:         `break`
13:     **end if**
14:     Set $b$ as the parent of $b$
15: **end while**
**Ensure:** $\mathcal{T}$

---

problems, we do not have to deal with permuting the dataset, thus significantly simplifying the algorithm and analysis.

In particular, upon the deletion request for user data $z_t$, we query the environment oracle for a *dummy* user $z'$, an arbitrary user data that has no correlation with the deleted user data. We then iterate through the path from leaf $t$ to the

root. For each node $b$ on the path, we update its clean value from $v_b$ to $v_b'$, to account for the fact that $x_t$ is replaced by $x_t'$. The involved part is how to update the noisy value $\tilde{v}_b$, as all the nodes that follow $b$ depend on the noisy value $\tilde{v}_b$, not the clean value $v_b$. Ideally, we want to re-use the old value of $\tilde{v}_b$ so that all its dependent nodes do not need to update. This is precisely the problem of designing a maximal coupling. This amounts to the rejection sampling step (Line 6-7) that checks if the old value of $\tilde{v}_b$ can be seen as a sample from $\mathcal{N}(v_b', \sigma^2 I)$. In particular, we compute the density ratio of the two Gaussians at the old value of $\tilde{v}_b$, i.e., $\frac{f_{\mathcal{N}(v_b', \sigma^2 I)}(\tilde{v}_b)}{f_{\mathcal{N}(v_b, \sigma^2 I)}(\tilde{v}_b)}$, and compare it against a random sample from a uniform distribution $\text{Unif}(0, 1)$. If it results in accept, move to the parent node of the current node and repeat. If it results in reject, reflect the sample (Line 9) (ensuring the new noisy value $\tilde{v}_b$ has correct distribution $\mathcal{N}(v_b', \sigma^2 I)$ by reflecting around the new mean) and retrain from scratch from the next leaf in the post-traversal order in the tree (Line 11).

*Remark* 3.1 (Space/time trade-off)*.* The binary tree requires $O(Td)$ space and $O(d \log T)$ per-episode updates, reasonable for systems that already log versioned episodic data; we make this trade-off explicit.

The key result in this section is the guarantee of exact unlearning and its relative computational complexity, as long as the environment oracle has a finite $\ell_2$ sensitivity. In particular, we assume the sensitivity of the environment oracle, imposing that changing a user data does not change the output of the environment oracle by too much.

**Assumption 3.2** ($\ell_2$ sensitivity)**.** There exists a constant $B > 0$ such that

$$\sup_{t, z, z', w_{1:t}} \|\mathbb{EO}_t(z; w_{1:t}) - \mathbb{EO}_t(z'; w_{1:t})\|_2 \leq B.$$

The nature of the following theorem is similar to (Ullah & Arora, 2023, Theorem 1), except that we focus on sequential problems rather than batch problems. Our proof is also significantly simplified, as, again, we do not have to deal with permuting the dataset (see Appendix B).

**Theorem 3.3.** *Fix any $\rho > 0$. Set $\sigma = \frac{B\sqrt{\log_2 T}}{\sqrt{2}\rho}$, where $B$ is the bound on sensitivity (Assumption 3.2). Then,*

1. *Algorithm 3 is exact unlearning w.r.t. Algorithm 2,*
2. *Algorithm 2 is $\rho$-TV-stable,*
3. *The probability of retraining in unlearning is $\rho\sqrt{2\log_2 T}$.*

**Proof Sketch (Theorem 1)**. The learning phase replaces raw prefix sums with correlated Gaussian noise via a binary-tree mechanism so that each query uses only $\mathcal{O}(\log T)$ shared noises. In unlearning, we traverse the path from the deleted leaf to the root and maximally couple the old

and new Gaussian nodes: for node $b$, we first "clean-update" its mean $v_b \mapsto v_{b'}$, then perform a rejection test that reuses $\tilde{v}_b$ whenever possible; on a failure we apply the reflection map $\tilde{v}_{b'} := v_{b'} + v_b - \tilde{v}_b$ (a Gaussian-to-Gaussian maximal coupling), and retrain only from the next post-order leaf. This yields exact unlearning by construction and confines retraining to rare coupling failures along $\mathcal{O}(\log T)$ nodes.

Our unlearning Algorithm 3 is a simplified instance of the batch tree-coupling framework in (Ullah & Arora, 2023): the sequential nature eliminates dataset permutation and reduces the coupling path to at most $\log T$ nodes per request, improving both exposition and constants. We now move on to applying the above unlearning framework to RL.

# 4. Reinforcement (Un)Learning

Designing an unlearning framework for reinforcement learning based on the general approach from Section 3 requires instantiating the environment oracle $\mathbb{EO}_t$ and defining the internal update function **UPDATE** for MDPs. After such instantiation, all the unlearning guarantees in Theorem 3.3 remain for RL, with a specific value of $B$ we discuss shortly. The key challenge here is, however, to design **UPDATE** for RL such that the RL algorithm has regret-optimal learning, despite being disturbed by injected noises for unlearning guarantees. In this section, we design **UPDATE**, built upon the famous UCB-VI (Azar et al., 2017), that attains nearly minimax-optimal regret-stability trade-offs.

**Realizing the Environment Oracle $\mathbb{EO}_t$ and the Summary Statistics $x_t$ for MDPs.** During episode $t$, the RL algorithm interacts with user $z_t$, generating an experience sequence $(s_1^t, a_1^t, r_1^t, \ldots, s_H^t, a_H^t, r_H^t)$. The environment oracle $\mathbb{EO}_t$ then returns summary statistics $x_t$ that encode this trajectory. In the MDP case, the summary statistics $x_t$ is defined as

$$x_t = \{x_t[h, s, a]\}_{(h,s,a)\in[H]\times\mathcal{S}\times\mathcal{A}}, \text{ where} \tag{1}$$

$x_t[h, s, a] = (\mathbb{1}_{\{(s_h^t, a_h^t)=(s,a)\}}, \mathbb{1}_{\{(s_h^t, a_h^t, s_{h+1}^t)=(s,a,s')\}}, r_h^t \cdot \mathbb{1}_{\{(s_h^t, a_h^t)=(s,a)\}})$. The dimension of the output from the environment oracle is $d = 2SAH + S^2AH$, corresponding to indicators for state-action occurrences, transitions, and associated rewards across the horizon.

**Sensitivity Parameter $B$ for MDPs.** Importantly, the sensitivity parameter of the environment oracle is bounded as $B \leq \sqrt{3H}$. This follows from the observation that, in any episode $t$, the generated statistic $x_t$ contains at most $3H$ non-zero components, one per time step for each of the indicator, transition, and reward terms, independent of the sizes of the state or action spaces.

The objective of this section is to design the internal update

function for RL, grounded in the statistics constructed above. Our proposed update function is presented in Algorithm 4.

Similar to the standard (non-stable) RL algorithm of Azar et al. (2017), Algorithm 4 follows the principle of optimistic value iteration, with adjustments to ensure stability through noise-perturbed statistics and confidence bonuses. In particular, let $N_h^t(s, a)$ and $N_h^t(s, a, s')$ denote the number of visits to the tuples $(h, s, a)$ and $(h, s, a, s')$, respectively, prior to episode $t$. Let $R_h^t(s, a)$ be the total reward accumulated at $(h, s, a)$ up to that point. Their noisy counterparts, denoted by $\widetilde{N}_h^t(s, a), \widetilde{N}_h^t(s, a, s'), \widetilde{R}_h^t(s, a)$, are computed using the binary tree mechanism described in Algorithm 2. For use in Algorithm 4, we define the following quantities:

$$\epsilon_{\text{app}}(\delta) := \sigma \ln T \left( 1 + \sqrt{2 \log(2T(S^2AH + 2SAH)/\delta)} \right),$$

$$\tilde{r}_h^t(s, a) := \frac{\widetilde{R}_h(s, a)}{\widetilde{N}_h^t(s, a)}, \widetilde{\mathbb{P}}_h^t(s'|s, a) := \frac{\widetilde{N}_h^t(s, a, s')}{\widetilde{N}_h^t(s, a)}$$

$$b(\tilde{n}) := (H+1)\sqrt{\frac{\ln(8SAHT/\delta)}{2(\tilde{n} - \epsilon_{\text{app}}(\frac{\delta}{4}))}} + \frac{\epsilon_{\text{app}}(\frac{\delta}{4})}{\tilde{n}}(1 + 2H(\sqrt{S} + 1)),$$

where the first term in the bonus function $b(\tilde{n})$ is a Hoeffding-style exploration bonus; the second term compensates for approximation error from noisy counts. During episode $t$, the algorithm performs value iteration using the stable estimators $\{\tilde{r}_h^t\}, \{\widetilde{\mathbb{P}}_h^t\}$ and the bonus function $b(\cdot)$, to compute a stable approximation $\widetilde{Q}^t$ of the Q-function. Intuitively, the first term of the bonus function accounts for the estimation error due to finite samples, while the second term compensates for the approximation error introduced by noisy (i.e., stabilized) counts. The algorithm then outputs greedy policy $\widetilde{\pi}^t$, which is used to generate action recommendations $a_{1:H}^t$ during episode $t$.

---

**Algorithm 4** UPDATE$(w_{1:t}, \tilde{u}_t)$

---

**Require:** $\tilde{u}_t = \{(\widetilde{N}_h^t(s, a), \widetilde{N}_h^t(s, a, s'), \widetilde{R}_h^t(s, a)) : (h, s, a) \in [H] \times \mathcal{S} \times \mathcal{A}\}$, bonus function $b : \mathbb{R} \to \mathbb{R}$, and approximation error $\epsilon_{\text{app}}(\frac{\delta}{4})$.
1: Initialize $Q_{H+1}^t(s, a) = H$
2: **for** $h = H, \ldots, 1$ **do**
3:    **if** $\widetilde{N}_h^t(s, a) \geq 2\epsilon_{\text{app}}(\frac{\delta}{4})$ **then**
4:       $\widetilde{Q}_h^t(s, a) = (\tilde{r}_h^t(s, a) + (\widetilde{\mathbb{P}}_h^t \widetilde{V}_{h+1}^t)(s, a) + b(\widetilde{N}_h^t(s, a))$
5:       $\widetilde{Q}_h^t(s, a) = \min\{H, \widetilde{Q}_h^t(s, a)\}$
6:    **else**
7:       $\widetilde{Q}_h^t(s, a) = H$
8:    **end if**
9:    $\forall(t, h), \widetilde{V}_h^t(s) = \max_{a \in \mathcal{A}} \widetilde{Q}_h^t(s, a)$ and $\widetilde{\pi}_h^t(s) = \arg\max_{a \in \mathcal{A}} \widetilde{Q}_h^t(s, a)$
10: **end for**
**Ensure:** $\widetilde{\pi}^t$

---

The main result of this section is the following regret bound for Algorithm 4.

**Theorem 4.1.** *Fix any $\rho > 0$ and set $\sigma = \frac{B\sqrt{\log_2 T}}{\sqrt{2}\rho}$. In Algorithm 2, instantiate the environment oracle $\mathbb{EO}_t$ (Line 3 of Algorithm 2) using Equation (1), and the algorithm oracle* UPDATE *(Line 6 of Algorithm 2) using Algorithm 4. Then, the resulting reinforcement learning algorithm is $\rho$-TV-stable. Furthermore, for any $\delta \in (0, 1)$, with probability at least $1 - \delta$, the regret of the algorithm satisfies: $R(T) \leq (H^3S^2A + H^2\sqrt{SAT} + H^{2.5}S^2A/\rho) \cdot \text{polylog}(T, H, S, A, 1/\delta)$.*

The regret bound comprises three terms: the first term captures the burn-in cost incurred before sublinear regret behavior emerges; the second term corresponds to the standard regret bound of the UCB-VI algorithm (Azar et al., 2017), using a Hoeffding-type bonus rather than a Bernstein-type bonus; and the third term accounts for the additional cost of stabilizing the RL algorithm with Gaussian noise injection. The complete proof is given in Appendix C.

*Remark* 4.2. If we set $\rho = \sqrt{\frac{HS^3A}{T}}$, then TV-stability does not incur any additional regret compared to the baseline algorithm. Moreover, with this choice of $\rho$, the probability of retraining, equal to $\rho\sqrt{\ln T}$, decreases at a sublinear rate as $T$ increases. However, since the cost of retraining grows with $T$ (e.g., $\mathcal{O}(T)$ for our RL baseline algorithm), maintaining the same order of regret as the non-TV-stable algorithm requires that the expected computational cost of unlearning scales as $\rho T = \sqrt{THS^3A}$.

*Remark* 4.3. The regret bound of $\rho$-TV-stable RL algorithms in Theorem 4.1 is structurally similar to the regret bounds established for $\epsilon$-Joint-Differential-Private (JDP) RL algorithms in (Vietri et al., 2020). Specifically, they show that the additional cost introduced by their proposed $\epsilon$-JDP algorithms is an additive term of order $\frac{H^3S^2A}{\epsilon}$. When setting $\rho = \epsilon$, our $\rho$-TV-stable algorithm incurs a lower additional cost, improving over the $\epsilon$-JDP bound by a factor of $\sqrt{H}$. This improvement is expected, as our framework uses Gaussian noise to ensure stability, whereas the JDP algorithms rely on Laplacian noise. Importantly, Gaussian noise yields a tighter sensitivity bound of $B = \sqrt{3H}$ compared to $B = 3H$ bound that would arise from using Laplace noise.

### 4.1. Minimax Lower Bounds

Next, we establish a minimax lower bound on the regret of any $\rho$-TV-stable RL algorithm. The detailed proof is given in Appendix D.

**Theorem 4.4.** *For any $\rho$-TV-stable RL algorithm, the following lower bound on regret holds:*

$$R(T) = \Omega\left(H\sqrt{SAT} + HSA/\rho\right).$$

This lower bound indicates that our upper bound is tight, up to a factor of $H$ in the first term and $H^{1.5}S$ in the second. The gap in the first term is reducible via sharper analysis, for example by using Bernstein-type bonus functions in place of Hoeffding-type bonuses, as used in our current design.

*Remark* 4.5. The lower bound in Theorem 4.4 matches, in both form and order, the lower bound established for $\rho$-JDP RL algorithms in (Vietri et al., 2020). However, it is important to note that lower bounds for DP algorithms do not directly imply lower bounds for TV-stable algorithms, since $\rho$-DP implies $2\rho$-TV-stability but there does not exist any function $f : \mathbb{R}_+ \to \mathbb{R}_+$ such that, $\rho$-TV-stability implies $f(\rho)$-DP (see Part 2 of Lemma A.5). Hence, (pure) DP is strictly stronger than TV stability, and thus our lower bound is strictly stronger. Despite this distinction, our proof follows a broadly similar framework to (Vietri et al., 2020), with a key technical deviation: we apply a different corollary (see Lemma A.6) of the foundational result of (Karwa & Vadhan, 2017, Lemma 6.1). Our proof is not only simpler and more intuitive, but also improves upon the analysis gap in (Vietri et al., 2020) by more cleanly demonstrating how any $\rho$-TV-stable RL algorithm can simulate a $\rho$-TV-stable multi-armed bandit algorithm (see Remark D.5).

**On the MDP Class.** Compared to the hard instances constructed by (Vietri et al., 2020), we use separate absorbing states per initial state; otherwise, an RL algorithm can learn that the shared "+" absorber dominates and bias actions for unseen initials, breaking the intended reduction to $S$ independent MABs and invalidating the lower bound. See Lemma D.4/Corollary D.3 for the S decoupled MAB simulation and the $HSA/\rho$ term in our bound.

## 5. Conclusion and Discussion

We presented the first theoretical and algorithmic framework for *provably exact unlearning* in RL, specifically within tabular Markov decision processes. Our framework leverages Total Variation (TV) stability to design sample-efficient, TV-stable RL algorithms that enable efficient unlearning. We established regret bounds that are nearly minimax-optimal for the class of TV-stable RL algorithms.

We emphasize that TV-stability is not necessary for unlearning in all frameworks. Algorithms lacking TV-stability may still permit alternative unlearning strategies, for example, through structural or compositional properties that enable bespoke procedures. Our theory thus characterizes a natural and rich subclass of unlearnable algorithms rather than all possible approaches. As the first results in RL to provide end-to-end guarantees combining regret-optimal learning with provable unlearning, we view this as a compelling foundational step forward.

Several important gaps remain in our understanding of exact unlearning for regret minimization and sequential learning

more broadly. First, there is still a nontrivial gap between our upper and lower bounds on regret for $\rho$-TV-stable RL algorithms. While the first term in the upper bound could potentially be tightened using sharper analysis (e.g., Bernstein-type bonuses in place of Hoeffding-type), it is unclear how to close the gap of $H^{1.5}S$ in the second term.

Second, our results are confined to the tabular setting. Achieving exact unlearning when the state space is large or continuous, necessitating function approximation, remains an open question, presenting both statistical and algorithmic challenges. The key insight behind our approach is maintaining sufficient statistics that support both learning and unlearning. In tabular MDPs, these take the form of visitation counts. In RL with function approximation (e.g., linear MDPs), one could maintain alternative sufficient statistics such as empirical covariance matrices or feature-based summaries. While we lack results in the general setting, this direction appears promising and may enable unlearning in more expressive models. Developing online-compatible, memory-efficient unlearning under function approximation remains an important direction for future work.

Third, exact unlearning, both in our work and in prior work (Ullah et al., 2021; Ullah & Arora, 2023), relies on data structures such as binary trees, which incur space complexity linear in the number of episodes $T$. While it is reasonable to expect that exact unlearning requires greater space than approximate unlearning, our understanding of the optimal space-time-utility tradeoffs in this regime remains limited. A fundamental question is whether it is possible to achieve exact unlearning with strong utility guarantees using only *sublinear* space complexity.

Fourth, while TV stability provides a natural and powerful mechanism for designing efficient exact unlearning algorithms, it remains an open question whether alternative mechanisms could yield improved trade-offs between utility, computational cost, and space complexity.

We hope our work lays the groundwork for addressing these open questions and inspires further research on the foundations of unlearning in sequential and interactive learning systems.

## Acknowledgements

This work was supported in part by NSF CAREER award IIS-1943251.

## Impact Statement

This paper presents work whose goal is to advance the field of Machine Learning. There are many potential societal consequences of our work, none which we feel must be specifically highlighted here.

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

## A. Technical Lemmas

**Lemma A.1.** *We have*

$$\int \min\{p(x), q(x)\}dx \geq 1 - \sqrt{\frac{1}{2}D_{KL}(P\|Q)},$$

*where $D_{KL}$ is the KL divergence. , and $p, q$ are the densities of $P$ and $Q$, respectively.*

*Proof.* We have

$$
\begin{aligned}
1 - \int \min\{p(x), q(x)\}dx &= \frac{1}{2}\int (p(x) + q(x) - 2\min\{p(x), q(x)\})dx \\
&= \frac{1}{2}\int |p(x) - q(x)|dx \\
&= \mathrm{TV}(P, Q) \\
&\leq \sqrt{\frac{1}{2}D_{\mathrm{KL}}(P\|Q)},
\end{aligned}
$$

where the last inequality is due to Pinsker's inequality. $\square$

**Lemma A.2.** *We have*

$$D_{KL}(\mathcal{N}(\mu, \sigma^2 I), \mathcal{N}(\nu, \sigma^2 I)) = \frac{\|\mu - \nu\|_2^2}{2\sigma^2}.$$

**Lemma A.3** (Concentration of spherical Gaussians)**.** *Let $X \sim \mathcal{N}(0, \sigma^2 I_d)$. For any $\delta$, with probability at least $1 - \delta$,*

$$\|X\|_2 \leq \sigma\sqrt{d}\left(1 + \sqrt{\frac{2\log(1/\delta)}{d}}\right).$$

**Lemma A.4** (Reflection coupling (Ullah et al., 2021, Lemma 1))**.** *Let $P$ and $Q$ be two distributions over $\mathbb{R}^d$. Let $\psi : \mathbb{R}^d \to \mathbb{R}^d$ be a bijection such that $f_P(\psi(x)) = f_Q(x)$ and $|det\frac{d\psi(x)}{dx}| = 1$, where $d\psi(x)/dx$ is the Jacobian of the multivariate map $\psi$. Let $x \sim P$ and $y = x$ if $Unif(0, 1) \leq \frac{f_Q(x)}{f_P(x)}$ and $y = \psi(x)$ otherwise. Then $(x, y)$ is a maximal coupling of $(P, Q)$.*

*In addition, two isotropic Gaussians $P, Q$ over $\mathbb{R}^d$ with means $\mu_P$ and $\mu_Q$ such that for any two vectors $x, y$, $f_P(x) = f_Q(y)$ if $\|\mu_P - x\| = \|\mu_Q - y\|$. Then, with the map $\psi(x) = \mu_Q + \mu_P - x$, $P, Q, \psi$ satisfy the conditions in the preceding result.*

### A.1. Connections to Different Stability Notions

Let $\underline{x} = (x_1, \ldots, x_n)$ be a dataset of $n$ elements where each $x_i \in \Omega$. Two datasets, $\underline{x}$ and $\underline{x}'$, both of size $n$, are called neighbors if they differ by one element.

*Definition* A.1 (Differential privacy (DP) (Dwork et al., 2006))**.** A randomized algorithm $M : \Omega^n \to \Omega_M$ is $(\epsilon, \delta)$ differentially private if for all neighboring datasets $\underline{x}, \underline{x}' \in \Omega^n$ and for all measurable sets of outputs $E \in \Omega_M$, we have

$$\Pr(M(\underline{x}) \in E) \leq e^\epsilon \Pr(M(\underline{x}') \in E) + \delta.$$

Note that $(\epsilon, 0)$-DP is called $\epsilon$-pure-DP.

*Definition* A.2 (TV-stability)**.** A randomized algorithm $M : \Omega^n \to \Omega_M$ is $\rho$-TV-stable if for all neighboring datasets $\underline{x}, \underline{x}' \in \Omega^n$, we have

$$\mathrm{TV}(\Pr(M(\underline{x})), \Pr(M(\underline{x}'))) \leq \rho.$$

**Lemma A.5** (Relationship between differential privacy and TV stability)**.** *We have*

1. *If a randomized algorithm $M$ is $(\epsilon, 0)$ differentially private, then $M$ is $(e^\epsilon - 1)$-TV-stable (which is $(2\epsilon)$-TV-stable, if $\epsilon \in [0, 1]$), but not vice versa, i.e., there exists no absolute constant $c > 0$ such that, if $M$ is $\epsilon$-TV-stable, then $M$ is $(c\epsilon, 0)$-DP.*

2. *TV-stability does not imply pure DP in the following strong sense: There does not exist any function $f : \mathbb{R}_+ \to \mathbb{R}_+$ such that, if $M$ is $\epsilon$-TV-stable for any $\epsilon \in (0, 1)$, then $M$ is $f(\epsilon)$-pure-DP.*

3. *A randomized algorithm $M$ is $(0, \delta)$ differentially private iff it is $\delta$-TV-stable.*

*Proof of Lemma A.5.* Let $P = \Pr(M(\underline{x}))$ and $Q = \Pr(M(\underline{x}'))$.

**For Part 1.** For any measurable event $E$, if $M$ is $(\epsilon, 0)$-DP, then $P(E) \leq e^\epsilon Q(E)$. Thus,

$$P(E) - Q(E) \leq (e^\epsilon - 1)Q \leq e^\epsilon - 1,$$

which implies that $\mathrm{TV}(P, Q) \leq e^\epsilon - 1$. For $\epsilon \in [0, 1]$, we have $e^\epsilon - 1 \leq 2\epsilon$.

**For Part 2.** Now suppose that $\mathrm{TV}(P, Q) \leq \epsilon$. We will construct a counterexample $(P, Q)$ where there does not exist any function $f : \mathbb{R}_+ \to \mathbb{R}_+$ such that $P(E) \leq e^{f(\epsilon)}Q(E), \forall E$, i.e., $P(E) - Q(E) \leq (e^{f(\epsilon)} - 1)Q(E)$. Assume there exists such $f$. Pick $P$ and $Q$ such that there is an event $E$ where $p - q = \epsilon$ and $q = \epsilon^{n+1}$, where $p = P(E), q = Q(E)$ (e.g., $P$ and $Q$ are supported on only two points). We have

$$\frac{q}{p - q}(e^{f(\epsilon)} - 1) = \epsilon^n(e^{f(\epsilon)} - 1) \overset{n \to \infty}{\to} 0,$$

Thus, we can pick $n$ such that $\frac{q}{p-q}(e^{f(\epsilon)} - 1) < 1$, leading to the contradiction.

**For Part 3.** It is trivial from the definition. $\qquad\square$

Part 1 implies that $\epsilon$-DP is stronger than TV-stability, thus a lower bound for $\epsilon$-DP algorithms does not trivially imply a lower bound for TV-stable algorithms.

The simple observation from Part 2 of Lemma A.5 turns the beautiful result of (Karwa & Vadhan, 2017, Lemma 6.1) from DP algorithms to TV-stable algorithms. In particular, the following lemma, a corollary of (Karwa & Vadhan, 2017, Lemma 6.1) and Part 2 of Lemma A.5, says that, if the TV distance between the two outputs of a $\rho$-TV-stable algorithm on an i.i.d. sample from two distributions $P$ and $Q$, respectively, scale with $\rho \cdot n \cdot \mathrm{TV}(P, Q)$ where $n$ is the sample size of the i.i.d. samples.

**Lemma A.6** ((Karwa & Vadhan, 2017, Lemma 6.1)). *For every pair of distribution $P$ and $Q$, every $\rho$-TV-stable randomized algorithm $M(x_{1:n})$, if $\mathbb{M}_P, \mathbb{M}_Q$ are the marginal distributions induced by running $M$ on the i.i.d. samples $x_{1:n}$ drawn from $P$ and $Q$, respectively, then*

$$TV(\mathbb{M}_P, \mathbb{M}_Q) \leq 4n\rho TV(P, Q).$$

The above lemma is a key to establish the lower bounds for regret of $\rho$-TV-stable RL algorithms in Appendix D.

## B. Unlearning Guarantees

### B.1. Proof of Part 1 of Theorem 3.3

*Proof of part 1 of Theorem 3.3.* This proof of part 1 mainly follows from the logic established in (Ullah & Arora, 2023, Lemma 4), only with significantly simplified steps and language – partly due to that we do not have to deal with permuting the dataset as in the original proof of (Ullah & Arora, 2023). The key idea is, again, to show that our unlearning algorithm is a construction of an approximately maximal coupling between the distributions of the output of the learning algorithm on the original data and the modified data, respectively.

Let $\mathcal{D} = \{z_1, \ldots, z_T\}$ and $\mathcal{D}' = \{z_1, \ldots, z_{t-1}, z', z_{t+1}, \ldots, z_T\}$, where $t$ is the position where the user data needs to be deleted. Let $P, Q$ be the probability over the range of the tree data structure induced by the output of the learning algorithm on $\mathcal{D}$ and $\mathcal{D}'$, respectively.

Let $\mathcal{T}, \mathcal{T}'$ be the binary trees constructed after running the learning algorithm on $\mathcal{D}$ and after unlearning, respectively. We order the nodes of a tree by the post-order traversal. That is, whenever we talk about any sense of orders of nodes in a tree, we implicitly mean the post-order traversal. For example, $\leq b$ denotes the set of all nodes that are prior to node $b$ in the post-order traversal. For any subset of nodes $S$, $\mathcal{T}_S$ denotes the sub-tree of $\mathcal{T}$ that contains only the nodes in $S$ and $Q_S$ denotes the marginal distribution of $Q$ on the nodes $S$.

The goal is to show that,

$$\Pr(\mathcal{T}' \in E) = Q(E), \forall \text{ measurable event } E. \tag{2}$$

Recall that each node $b$ in a tree consists of clean value $v_b$, noisy value $\tilde{v}_b$ and a model $w$ (if the node is a leaf). The clean value $v_b$ and the model $w$ are outputs of functions on only the noisy prefix sum of the nodes that precede $b$ (in the post-traversal order), while the noisy value $\tilde{v}_b$ is a noisy version of $v_b$, added with an independent noise. Thus, to prove Equation (2), it suffices to prove that,

$$\Pr(\widetilde{\mathcal{T}}' \in E) = \widetilde{Q}(E), \forall \text{ measurable event } E. \tag{3}$$

where $\widetilde{\mathcal{T}}'$ is the same as $\widetilde{\mathcal{T}}$ except each node in $\widetilde{\mathcal{T}}'$ only represents the noisy value, and $\widetilde{Q}$ is the marginal distribution of $Q$ on the noisy values.

We will prove Equation (3) by induction by $k$,

$$\Pr(\widetilde{\mathcal{T}}'_{\leq k} \in E) = \widetilde{Q}_{\leq k}(E), \forall \text{ measurable event } E. \tag{4}$$

**Base case $k = 1$.** If $t > 1$, we have $\Pr(\widetilde{\mathcal{T}}'_{\leq 1} \in E) = \tilde{Q}_{\leq 1}(E)$, since the deleted node $t$ does not depend on node $k = 1$. Consider the case $t = 1$. In this case, (Ullah et al., 2021, Lemma 1) implies that $\Pr(\tilde{v}_1 \in E) = \tilde{Q}_{\leq 1}(E)$.

**Induction step.** We now proceed to the induction step: Let us assume that Equation (4) holds for all nodes up to $k$, we will prove that it holds for node $k + 1$ as well. If the deleted position $t > k + 1$, Equation (4) holds for $k + 1$, since all nodes $< t$ do not get affected by changing node $t$.

We only need to consider $t \leq k + 1$. For this case, there are only the following further cases:

- A: All rejection sampling steps prior to node $k + 1$ resulted in accept.
    - AP: Node $k + 1$ is on the path from node $t$ to the root
        * APA: The rejection sampling step at node $k + 1$ resulted in accept.
        * APR: The rejection sampling step at node $k + 1$ resulted in reject.
    - AN: Node $k + 1$ is not on the path from node $t$ to the root

- R: Some rejection sampling steps prior to node $k + 1$ resulted in reject.
    - RP: Node $k + 1$ is on the path from node $t$ to the root
    - RN: Node $k + 1$ is not on the path from node $t$ to the root

For case RP, node $k + 1$ is not on the nodes that got re-trained from scratch as we retrained the subtree starting from the leaf that is right next after node $k + 1$. But $v_{k+1}$ still got updated to ensure that the clean value of node $k + 1$ is the sum of the clean values of its children. Thus, we have $\Pr(\tilde{v}_{k+1} \in E_{k+1} | \text{RP}, \widetilde{\mathcal{T}}'_{\leq k} \in E_{\leq k}) = \widetilde{Q}_{k+1}(E_{k+1} | E_{\leq k})$.

For case RN, node $k + 1$ got re-trained from scratch, thus we must have $\Pr(\tilde{v}_{k+1} \in E_{k+1} | \text{RN}, \widetilde{\mathcal{T}}'_{\leq k} \in E_{\leq k}) = \widetilde{Q}_{k+1}(E_{k+1} | E_{\leq k})$.

For case AN, we have $\Pr(\tilde{v}_{k+1} \in E_{k+1}|\mathrm{AN}, \widetilde{\mathcal{T}}'_{\leq k} \in E_{\leq k}) = \widetilde{Q}_{k+1}(E_{k+1}|E_{\leq k})$, as the clean value of node $k+1$ does not get modified during unlearning when node $t$ is deleted.

For case AP, we also have $\Pr(\tilde{v}_{k+1} \in E_{k+1}|\mathrm{AP}, \widetilde{\mathcal{T}}_{\leq k} \in E_{\leq k}) = \widetilde{Q}_{k+1}(E_{k+1}|E_{\leq k})$, as the sub-tree associated with nodes $\leq k$ get unmodified during unlearning in case A, and the rejection sampling and the reflection map at node $k+1$ results in a valid coupling ((Ullah et al., 2021, Lemma 1)).

Thus, we must have $\Pr(\tilde{v}_{k+1} \in E_{k+1}|\widetilde{\mathcal{T}}'_{\leq k} \in E_{\leq k}) = \widetilde{Q}_{k+1}(E_{k+1}|E_{\leq k})$. Overall, we have

$$\begin{aligned}
\Pr(\widetilde{\mathcal{T}}'_{\leq k+1} \in E_{\leq k+1}) &= \Pr(\tilde{v}_{k+1} \in E_{k+1}|\widetilde{\mathcal{T}}'_{\leq k} \in E_{\leq k}) \Pr(\widetilde{\mathcal{T}}'_{\leq k} \in E_{\leq k}) \\
&= \widetilde{Q}_{k+1}(E_{k+1}|E_{\leq k})\widetilde{Q}_{\leq k}(E_{\leq k}) \\
&= \widetilde{Q}_{\leq k+1}(E_{\leq k+1}),
\end{aligned}$$

where $E_{\leq k+1} = E_1 \times \ldots \times E_{k+1}$, and where the second equation uses the induction assumption that $\Pr(\widetilde{\mathcal{T}}'_{\leq k} \in E_{\leq k}) = \widetilde{Q}_{\leq k}(E_{\leq k})$. $\qquad\square$

## B.2. Proof of Part 2 of Theorem 3.3

*Definition* B.1. A mechanism $\mathcal{A}$ is $(\alpha, \epsilon(\alpha))$-RDP if for any two datasets $S$ and $S'$ that differ by one entry, we have

$$D_\alpha(\mathcal{S}\|\mathcal{S}') \leq \epsilon(\alpha)$$

RDP (Mironov, 2017) is proposed as a notion of DP that complements the weakness of the standard approximate DP (e.g., never compromise a total breach of privacy as in $(\epsilon, \delta)$-DP). RDP implies approximate DP but not the reverse. So in terms of weaknesses, $DP < RDP < ADP$.

**Lemma B.1.** *(RDP implies TV stability (Ullah & Arora, 2023)) If an algorithm satisfies $(\alpha, \epsilon(\alpha))$-RDP, then it satisfies $\sqrt{\lim_{\alpha \to 1} \epsilon(\alpha)}$-TV stability.*

**Lemma B.2** ((Mironov, 2017)). *Let $f$ be a real-valued function with $\ell_2$-sensitivity of $B$. The gaussian mechanism of adding $\mathcal{N}(0, \sigma^2)$ is $(\alpha, \alpha B^2/(2\sigma^2))$-RDP for any $\alpha \geq 1$.*

*Proof of part 2 of Theorem 3.3.* The proof is standard following the techniques in differential privacy with Gaussian noises and the two lemmas above. In particular, we will compute the RDP of the binary tree before and after deleting one user data. When we change one user data, there are at most $\log_2 T$ nodes in the binary tree that get impacted. Each impacted node is $(\alpha, \alpha B^2/(2\sigma^2))$-RDP, by Lemma B.2. Using the composition theorem (Mironov, 2017, Proposition 1), the impacted path is $(\alpha, \alpha B^2 \log_2 T/(2\sigma^2))$-RDP, as there are at most $\log_2 T$ nodes in the path. Since the other nodes do not get impacted, the entire tree is $(\alpha, \alpha B^2 \log_2 T/(2\sigma^2))$-RDP. Thus, by Lemma B.1, the entire tree is $\frac{B\sqrt{\log_2 T}}{\sqrt{2}\sigma}$-TV-stable. $\qquad\square$

## B.3. Proof of Part 3 of Theorem 3.3

*Proof of part 3 of Theorem 3.3.* During unlearning, re-training is only triggered when a rejection sampling step fails at some node $b$ in the path from the node $t$ to the root. Let us define this path by $\mathtt{path} = \{b_1, \ldots b_l\}$, in the order from the leaf node $t = b_1$ to the root $b_l$, where $l = |\mathtt{path}|$. Let $\mathtt{Accept}$ be the event where the rejection sampling steps in all nodes

in `path` succeed. Let $\zeta = (\zeta_1, \ldots, \zeta_l) \sim \text{Unif}(0,1)^l$. We have,

$$
\begin{aligned}
\Pr(\texttt{Accept}) &= \mathbb{E}_{\mathcal{T},\zeta} \prod_{b \in \texttt{path}} 1\left\{ \zeta_b \le \frac{f_{\mathcal{N}(v_b', \sigma^2 I)}(\tilde{v}_b)}{f_{\mathcal{N}(v_b, \sigma^2 I)}(\tilde{v}_b)} \right\} \\
&= \mathbb{E}_{\mathcal{T}} \mathbb{E}_{\zeta|\mathcal{T}} \left[ \prod_{b \in \texttt{path}} 1\left\{ \zeta_b \le \frac{f_{\mathcal{N}(v_b', \sigma^2 I)}(\tilde{v}_b)}{f_{\mathcal{N}(v_b, \sigma^2 I)}(\tilde{v}_b)} \right\} \right] \\
&= \mathbb{E}_{\mathcal{T}} \left[ \prod_{b \in \texttt{path}} \min\left\{ 1, \frac{f_{\mathcal{N}(v_b', \sigma^2 I)}(\tilde{v}_b)}{f_{\mathcal{N}(v_b, \sigma^2 I)}(\tilde{v}_b)} \right\} \right] \\
&= \mathbb{E}_{\mathcal{T}} \left[ \prod_{b \in \texttt{path}} \min\left\{ f_{\mathcal{N}(v_b, \sigma^2 I)}(\tilde{v}_b), f_{\mathcal{N}(v_b', \sigma^2 I)}(\tilde{v}_b) \right\} \right] \\
&= \int_{v_{b_1}, \tilde{v}_{b_1}} \cdots \int_{v_{b_l}, \tilde{v}_{b_l}} \prod_{i=1}^{l} \min\left\{ 1, \frac{f_{\mathcal{N}(v_{b_i}', \sigma^2 I)}(\tilde{v}_{b_i})}{f_{\mathcal{N}(v_{b_i}, \sigma^2 I)}(\tilde{v}_{b_i})} \right\} dP(v_{b_l}, \tilde{v}_{b_l} | v_{b \le l-1}, \tilde{v}_{b \le l-1}) \ldots dP(v_{b_1}, \tilde{v}_{b_1}) \\
&= \int_{v_{b_1}, \tilde{v}_{b_1}} \cdots \int_{v_{b_l}, \tilde{v}_{b_l}} \prod_{i=1}^{l} \min\left\{ 1, \frac{f_{\mathcal{N}(v_{b_i}', \sigma^2 I)}(\tilde{v}_{b_i})}{f_{\mathcal{N}(v_{b_i}, \sigma^2 I)}(\tilde{v}_{b_i})} \right\} f_{\mathcal{N}(v_{b_l}, \sigma^2 I)}(\tilde{v}_{b_l}) dP(v_{b_l}, | v_{b \le l-1}, \tilde{v}_{b \le l-1}) \ldots dP(v_{b_1}, \tilde{v}_{b_1}) \\
&\ge \prod_{i=1}^{l} (1 - \frac{B}{\sigma}) \\
&\ge 1 - l \frac{B}{\sigma} \\
&\ge 1 - \frac{B \log T}{\sigma}
\end{aligned}
$$

where the third equality follows as $\zeta$ is independent of $\mathcal{T}$, the second inequality is due to Ullah & Arora (2023, Lemma 7), the last inequality is due to the fact that the maximum length of the path from a leaf to the root is $\log T$, and the first inequality is due to the following inequality (and induction from $l$ to 1):

$$
\begin{aligned}
\int \min\left\{ 1, \frac{f_{\mathcal{N}(v_{b_i}', \sigma^2 I)}(\tilde{v}_{b_i})}{f_{\mathcal{N}(v_{b_i}, \sigma^2 I)}(\tilde{v}_{b_i})} \right\} f_{\mathcal{N}(v_{b_l}, \sigma^2 I)}(\tilde{v}_{b_l}) dP(\tilde{v}_{b_l} | v_{b_l}) &= \int \min\left\{ f_{\mathcal{N}(v_{b_i}, \sigma^2 I)}(\tilde{v}_{b_i}), f_{\mathcal{N}(v_{b_i}', \sigma^2 I)}(\tilde{v}_{b_i}) \right\} dP(\tilde{v}_{b_l} | v_{b_l}) \\
&\ge 1 - \frac{1}{2} \sqrt{\frac{\|v_{b_i}' - v_{b_i}\|_2^2}{\sigma^2}} \\
&\ge 1 - \frac{B}{\sigma},
\end{aligned}
$$

where the first inequality follows from Lemma A.1 and Lemma A.2, and the last inequality follows from that $\|v_{b_i}' - v_{b_i}\|_2^2 = \|x_t - x_t'\|_2^2 \le B^2$, by Assumption 3.2. $\qquad\square$

## C. Regret Bounds

**Notations.** Let $N_h^t(s,a)$ and $N_h^t(s,a,s')$ be the number of visits to $(h,s,a)$ and $(h,s,a,s')$, respectively, right before the start of episode $t$. Let $R_h^t(s,a)$ be the total rewards collected at $(h,s,a)$ right before the start of episode $t$. Let $\widetilde{N}_h^t(s,a), \widetilde{N}_h^t(s,a,s'), \widetilde{R}_h^t(s,a)$ be the noisy variants of $N_h^t(s,a), N_h^t(s,a,s'), R_h^t(s,a)$, respectively, obtained from the binary mechanism in Algorithm 3.

Let us define the following quantities:

$$\widehat{\mathbb{P}}_h^t(s'|s,a) := \frac{N_h^t(s,a,s')}{N_h^t(s,a)},$$

$$\widetilde{\mathbb{P}}_h^t(s'|s,a) := \frac{\widetilde{N}_h^t(s,a,s')}{\widetilde{N}_h^t(s,a)},$$

$$\hat{r}_h^t(s,a) := \frac{R_h^t(s,a)}{N_h^t(s,a)},$$

$$\tilde{r}_h^t(s,a) := \frac{\widetilde{R}_h^t(s,a)}{\widetilde{N}_h^t(s,a)},$$

$$\epsilon_{\text{app}}(\delta) := \sigma \ln T \left(1 + \sqrt{2 \log(2T(S^2 AH + 2SAH)/\delta)}\right)$$

$$\Delta_h^t := \widetilde{V}_h^t(s_h^t) - V_h^{\widetilde{\pi}^t}(s_h^t)$$

$$\zeta_h^t := [\mathbb{P}_h(\widetilde{V}_{h+1}^t - V_{h+1}^{\widetilde{\pi}^t})](s_h^t, a_h^t) - \Delta_{h+1}^t$$

## C.1. Basic Lemmas

**Lemma C.1.** *Fix any $\delta$. With probability at least $1 - \delta/4$, for all $(t,h,s,a) \in [T] \times [H] \times \mathcal{S} \times \mathcal{A}$ and all $V : \mathcal{S} \times \mathcal{A} \to [-H, H]$, we have*

1. $|\widetilde{N}_h^t(s,a) - N_h^t(s,a)| \leq \epsilon_{app}(\delta/4)$,

2. $|\tilde{r}_h^t(s,a) - \hat{r}_h^t(s,a)| \leq \frac{\epsilon_{app}(\delta/4)}{|\widetilde{N}_h^t(s,a)|}$,

3. $|[(\widetilde{\mathbb{P}}_h^t - \widehat{\mathbb{P}}_h)V](s,a)| \leq \frac{\epsilon_{app}(\delta/4)}{|\widetilde{N}_h^t(s,a)|} H(S+1)$.

*Proof of Lemma C.1.* The proof directly follows from the fact that every noisy prefix sum $\widetilde{N}_h^t(s,a)$ for a fixed $(h,s,a)$ is the result of perturbing $N_h^t(s,a)$ with at most $\log T$ independent spherical noise sampled from $\mathcal{N}(0, \sigma^2)$. The final form of the lemma simply follows from the concentration of spherical Gaussians (Lemma A.3) and the union bound over (at most) $2T$ nodes and $HS^2 A + 2HSA$ dimensions in each $u_t$ in the tree. $\square$

**Lemma C.2** (Hoeffding's inequality)**.** *Fix any $\delta > 0$. With probability at least $1 - \delta/4$, for all $(t,h,s,a) \in [T] \times [H] \times \mathcal{S} \times \mathcal{A}$, we have*

$$|\hat{r}_h^t(s,a) - r_h(s,a)| \leq \sqrt{\frac{\ln(8SAHT/\delta)}{2N_h^t(s,a)}},$$

$$[(\widehat{\mathbb{P}}_h^t - \mathbb{P}_h)V_{h+1}^*](s,a) \leq H\sqrt{\frac{\ln(8SAHT/\delta)}{2N_h^t(s,a)}}.$$

**Lemma C.3.** *Fix any $\delta > 0$. With probability at least $1 - \delta/4$, for all $(t,h,s,a,s')$:*

$$\widehat{\mathbb{P}}_h^t(s'|s,a) - \mathbb{P}_h(s'|s,a) \leq (8H + \frac{1}{3})\frac{\ln(4THS^2 A/\delta)}{N_h^t(s,a)} + \frac{1}{H}\mathbb{P}_h(s'|s,a).$$

*Proof of Lemma C.3.* By Bernstein's inequality, for any fixed $t, h, s, a, s', \delta$, with probability at least $1 - \delta$, we have

$$\widehat{\mathbb{P}}_h^t(s'|s,a) - \mathbb{P}_h(s'|s,a) < \frac{\ln(1/\delta)}{3N_h^t(s,a)} + \sqrt{\frac{2\mathbb{P}_h(s'|s,a)\ln(1/\delta)}{N_h^t(s,a)}}$$

$$\leq (8H + \frac{1}{3})\frac{\ln(1/\delta)}{N_h^t(s,a)} + \frac{1}{H}\mathbb{P}_h(s'|s,a),$$

where the last inequality uses Cauchy-Schwartz inequality. Using the union bound and rescaling $\delta$ completes our proof. $\square$

**Lemma C.4.** *With probability at least $1 - \delta/4$, for all $h \in [H]$, we have*

$$\sum_{t=1}^{T} \zeta_h^t \leq H \sqrt{32T \ln(4H/\delta)}.$$

*Proof of Lemma C.4.* For any fixed $h$, $\{\zeta_h^t\}_{t \in [T]}$ is a martingale difference sequence. Applying Azuma inequality completes our proof. $\square$

**Lemma C.5.** *Let $I_2 = \{t \in [T] : \widetilde{N}_h^t(s_h^t, a_h^t) \geq 2\epsilon_{app}(\delta/4), \forall h \in [H]\}$ and $I_1 = [T] \backslash I_2$. If the inequalities in Lemma C.1 hold, we have*

$$|I_1| \leq 3HSA\epsilon_{app}(\delta/4).$$

*Proof of Lemma C.5.* Note that

$$I_1 = \{t \in [T] : \exists h \in [H], \widetilde{N}_h^t(s_h^t, a_h^t) < 2\epsilon_{\text{app}}(\delta/4)\}.$$

Let us define

$$I_1' = \{t \in [T] : \exists h \in [H], N_h^t(s_h^t, a_h^t) < 3\epsilon_{\text{app}}(\delta/4)\}.$$

If the inequalities in Lemma C.1 hold, then $I_1 \subseteq I_1'$. Thus, it suffices to bound $|I_1'|$. Each $t \in I_1'$ must create at least one mapping to $[H] \times \mathcal{S} \times \mathcal{A}$ via $(h, s, a)$ such that $N_h^t(s, a) < 3\epsilon_{\text{app}}(\delta/4)$ and $(s, a) = (s_h^t, a_h^t)$. For any $(h, s, a)$, there are at most $3\epsilon_{\text{app}}(\delta/4)$ elements in $I_1'$ that create a mapping to $(h, s, a)$, because otherwise $(h, s, a)$ must be visited at least $3\epsilon_{\text{app}}(\delta/4)$ by some $t \in I_1'$, leading to a contradiction. Overall, the number of elements in $I_1'$ must be at most $3\epsilon_{\text{app}}(\delta/4)HSA$. $\square$

*Remark* C.6 (Intuition for Lemma C.5). Since $N_h^t(s, a)$ counts visits to $(h, s, a)$ before episode $t$ and increases by 1 each time $(h, s, a)$ is visited, the $k$-th episode to visit a fixed $(h, s, a)$ has $N_h^t(s, a) \geq k - 1$. For this episode to be in $I_1'$ requires $k - 1 < 3\epsilon_{\text{app}}(\delta/4)$, so at most $\lfloor 3\epsilon_{\text{app}}(\delta/4) \rfloor$ episodes visiting any fixed $(h, s, a)$ can be in $I_1'$. Summing over $HSA$ tuples gives $|I_1'| \leq 3HSA\epsilon_{\text{app}}(\delta/4)$.

## C.2. Proof of the Regret Bound

**Lemma C.7** (Optimism). *Fix any $\delta > 0$. If we set the bonus function as follows:*

$$b(\tilde{n}) = (H + 1)\sqrt{\frac{\ln(8SAHT/\delta)}{2(\tilde{n} - \epsilon_{app}(\delta/4))}} + \frac{\epsilon_{app}(\delta/4)}{\tilde{n}}(1 + 2H(\sqrt{S} + 1)),$$

*then, with probability at least $1 - \delta/2$, for all $(t, h, s, a) \in [T] \times [H] \times \mathcal{S} \times \mathcal{A}$, we have*

$$\widetilde{Q}_h^t(s, a) \geq Q_h^*(s, a) \text{ and } \widetilde{V}_h^t(s) \geq V_h^*(s), \forall(h, s, a). \tag{5}$$

*Proof of Lemma C.7.* We use induction to prove the lemma. For a fixed episode $t$, consider $h = H + 1, H, \ldots, 1$. The inequalities hold for $h = H + 1$. Assume $\widetilde{V}_{h+1}^t(s) \geq V_{h+1}^*(s)$ for all $s$ and some $h$. If the algorithm has never visited $(h, s, a)$ prior to episode $t$, then $\widetilde{Q}_h^t(s, a)$ has never got updated; thus, $\widetilde{Q}_h^t(s, a) = H \geq Q_h^*(s, a)$. Otherwise, we have

$$\widetilde{Q}_h^t(s, a) = \begin{cases} \min\left\{H, \tilde{r}_h(s, a) + (\widetilde{\mathbb{P}}_h^t \widetilde{V}_{h+1}^t)(s, a) + b(\widetilde{N}_h^t(s, a))\right\} & \text{if } \widetilde{N}_h^t(s, a) \geq 2\epsilon_{\text{app}}(\delta/4) \\ H & \text{if } \widetilde{N}_h^t(s, a) < 2\epsilon_{\text{app}}(\delta/4). \end{cases}$$

If $\widetilde{N}_h^t(s,a) < 2\epsilon_{\text{app}}(\delta/4)$ or $\widetilde{N}_h^t(s,a) \geq 2\epsilon_{\text{app}}(\delta/4)$ and the minimum is $H$, then $\widetilde{Q}_h^t(s,a) = H \geq Q_h^*(s,a)$. We only need to consider the case $\widetilde{N}_h^t(s,a) \geq 2\epsilon_{\text{app}}(\delta/4)$ and the minimum is not $H$. Then we have

$$
\begin{aligned}
\widetilde{Q}_h^t(s,a) - Q_h^*(s,a) &= \tilde{r}_h(s,a) - r_h(s,a) + (\widetilde{\mathbb{P}}_h^t \widetilde{V}_{h+1}^t)(s,a) - (\mathbb{P}_h V_{h+1}^*)(s,a) + b(\widetilde{N}_h^t(s,a)) \\
&= \tilde{r}_h^t(s,a) - r_h(s,a) + [(\widetilde{\mathbb{P}}_h^t - \mathbb{P}_h)V_{h+1}^*](s,a) + \widetilde{\mathbb{P}}_h^t(\widetilde{V}_{h+1}^t - V_{h+1}^*) + b(\widetilde{N}_h^t(s,a)) \\
&= \hat{r}_h^t(s,a) - r_h(s,a) + [(\widehat{\mathbb{P}}_h^t - \mathbb{P}_h)V_{h+1}^*](s,a) + \widehat{\mathbb{P}}_h^t(\widetilde{V}_{h+1}^t - V_{h+1}^*) \\
&\quad + \tilde{r}_h^t(s,a) - \hat{r}_h^t(s,a) + [(\widetilde{\mathbb{P}}_h^t - \widehat{\mathbb{P}}_h^t)V_{h+1}^*](s,a) + (\widetilde{\mathbb{P}}_h^t - \widehat{\mathbb{P}}_h^t)(\widetilde{V}_{h+1}^t - V_{h+1}^*) + b(\widetilde{N}_h^t(s,a)) \\
&\geq -(H+1)\sqrt{\frac{\ln(8SAHT/\delta)}{2N_h^t(s,a)}} - \frac{\epsilon_{\text{app}}(\delta/4)}{|\widetilde{N}_h^t(s,a)|}(1 + 2H(S+1)) + b(\widetilde{N}_h^t(s,a)) \\
&\geq 0
\end{aligned}
$$

where the first inequality is due to Lemma C.2 and Lemma C.1, and the last inequality is due to the definition of the bonus function. and Lemma C.1 and the condition $\widetilde{N}_h^t(s,a) \geq 2\epsilon_{\text{app}}(\delta/4)$. $\qquad\square$

**Regret analysis.** We are now ready to give a full proof for part 3 of Theorem 4.1.

*Proof of part 3 of Theorem 4.1.* Let $E$ be the event that, Equation (5) and all the inequalities in Lemma C.1, Lemma C.2, and Lemma C.3 hold simultaneously. We have $\Pr(E) \geq 1 - \delta$. Let $I_2 = \{t \in [T] : \widetilde{N}_h^t(s_h^t, a_h^t) \geq 2\epsilon_{\text{app}}(\delta/4), \forall h \in [H]\}$ and $I_1 = [T] \backslash I_2$.

Under event $E$, we have

$$
\text{Regret}(T) = \sum_{t=1}^T V_1^*(s_1^t) - V_1^{\widetilde{\pi}^t}(s_1^t) \leq \sum_{t=1}^T \widetilde{V}_1^t(s_1^t) - V_1^{\widetilde{\pi}^t}(s_1^t) = \sum_{t=1}^T \Delta_1^t,
$$

due to Lemma C.7.

Observe that action $a_h^t = \widetilde{\pi}_h^t(s_h^t) = \arg\max_{a \in \mathcal{A}} \widetilde{Q}_h^t(s_h^t, a)$. Thus,

$$
\Delta_h^t = \widetilde{V}_h^t(s_h^t) - V_h^{\widetilde{\pi}^t}(s_h^t) = \widetilde{Q}_h^t(s_h^t, a_h^t) - Q_h^{\widetilde{\pi}^t}(s_h^t, a_h^t).
$$

For any $t \in I_2$, we have

$$
\widetilde{Q}_h(s^t, a_h^t) - Q_h^{\widetilde{\pi}^t}(s_h^t, a_h^t) \leq \tilde{r}_h^t(s_h^t, a_h^t) - r_h(s_h^t, a_h^t) + (\widetilde{\mathbb{P}}_h^t \widetilde{V}_{h+1}^t - \mathbb{P}_h V_{h+1}^{\widetilde{\pi}^t})(s_h^t, a_h^t) + b_h^t(\widetilde{N}_h^t(s_h^t, a_h^t)).
$$

Under event $E$, for all $t, h$, we have

$$(\widetilde{\mathbb{P}}_h^t \widetilde{V}_{h+1}^t - \mathbb{P}_h V_{h+1}^{\widetilde{\pi}^t})(s_h^t, a_h^t)$$
$$= [(\widetilde{\mathbb{P}}_h^t - \mathbb{P}_h)V_{h+1}^*](s_h^t, a_h^t) + [(\widetilde{\mathbb{P}}_h^t - \mathbb{P}_h)(\widetilde{V}_{h+1}^t - V_{h+1}^*)](s_h^t, a_h^t) + [\mathbb{P}_h(\widetilde{V}_{h+1}^t - V_{h+1}^{\widetilde{\pi}^t})](s_h^t, a_h^t)$$
$$= [(\widetilde{\mathbb{P}}_h^t - \mathbb{P}_h)V_{h+1}^*](s_h^t, a_h^t) + [(\widetilde{\mathbb{P}}_h^t - \widehat{\mathbb{P}}_h)(\widetilde{V}_{h+1}^t - V_{h+1}^*)](s_h^t, a_h^t)$$
$$+ [(\widehat{\mathbb{P}}_h^t - \mathbb{P}_h)(\widetilde{V}_{h+1}^t - V_{h+1}^*)](s_h^t, a_h^t) + [\mathbb{P}_h(\widetilde{V}_{h+1}^t - V_{h+1}^{\widetilde{\pi}^t})](s_h^t, a_h^t)$$
$$\leq H\sqrt{\frac{\ln(8SAHT/\delta)}{2N_h^t(s, a)}} + \frac{\epsilon_{\text{app}}(\delta/4)}{\widetilde{N}_h^t(s, a)}H(S+1)$$
$$+ [(\widehat{\mathbb{P}}_h^t - \mathbb{P}_h)(\widetilde{V}_{h+1}^t - V_{h+1}^*)](s_h^t, a_h^t) + [\mathbb{P}_h(\widetilde{V}_{h+1}^t - V_{h+1}^{\widetilde{\pi}^t})](s_h^t, a_h^t)$$
$$\leq H\sqrt{\frac{\ln(8SAHT/\delta)}{2N_h^t(s_h^t, a_h^t)}} + \frac{\epsilon_{\text{app}}(\delta/4)}{\widetilde{N}_h^t(s, a)}H(S+1)$$
$$+ SH(8H + \frac{1}{3})\frac{\ln(THS^2A/\delta)}{N_h^t(s_h^t, a_h^t)} + \frac{1}{H}[\mathbb{P}_h(\widetilde{V}_{h+1}^t - V_{h+1}^*)](s_h^t, a_h^t) + [\mathbb{P}_h(\widetilde{V}_{h+1}^t - V_{h+1}^{\widetilde{\pi}^t})](s_h^t, a_h^t)$$
$$\leq H\sqrt{\frac{\ln(8SAHT/\delta)}{2N_h^t(s, a)}} + \frac{\epsilon_{\text{app}}(\delta/4)}{\widetilde{N}_h^t(s_h^t, a_h^t)}H(S+1)$$
$$+ SH(8H + \frac{1}{3})\frac{\ln(8THS^2A/\delta)}{N_h^t(s_h^t, a_h^t)} + (1 + \frac{1}{H})[\mathbb{P}_h(\widetilde{V}_{h+1}^t - V_{h+1}^{\widetilde{\pi}})](s_h^t, a_h^t)$$

where the first inequality is due to Lemma C.2 and the part 3 of Lemma C.1, the second inequality is due to $\widetilde{V}_{h+1}^t \geq V_{h+1}^*$ (Lemma C.7) and Lemma C.3, and the last inequality uses $V_{h+1}^* \geq V_{h+1}^{\widetilde{\pi}^t}$.

In addition, under event $E$, for all $t, h$, we have

$$\widetilde{r}_h^t(s_h^t, a_h^t) - r_h(s_h^t, a_h^t) \leq H\sqrt{\frac{\ln(8SAHT/\delta)}{2N_h^t(s_h^t, a_h^t)}} + \frac{\epsilon_{\text{app}}(\delta/4)}{|\widetilde{N}_h^t(s_h^t, a_h^t)|}.$$

Overall, we have that, under event $E$, for all $(t, h) \in I_2 \times [H]$,

$$\Delta_h^t \leq \underbrace{2H\sqrt{\frac{\ln(8SAHT/\delta)}{2N_h^t(s_h^t, a_h^t)}} + \frac{\epsilon_{\text{app}}(\delta/4)}{\widetilde{N}_h^t(s_h^t, a_h^t)}(1 + H(S+1)) + SH(8H + \frac{1}{3})\frac{\ln(8THS^2A/\delta)}{N_h^t(s_h^t, a_h^t)} + b_h^t(\widetilde{N}_h^t(s_h^t, a_h^t))}_{\xi_h^t}$$
$$+ (1 + \frac{1}{H})(\zeta_h^t + \Delta_{h+1}^t).$$

By recursion over $h = 1, \ldots, H$, the above inequality implies that

$$\Delta_1^t \leq \sum_{h=1}^H (1 + \frac{1}{H})^{h-1}\xi_h^t + e\sum_{h=1}^H (1 + \frac{1}{H})^h \zeta_h^t$$
$$\leq e\sum_{h=1}^H \xi_h^t + e\sum_{h=1}^H \zeta_h^t.$$

Thus, under event $E$, we have

$$\sum_{t \in I_2} \Delta_1^t \leq e\sum_{t \in I_2}\sum_{h=1}^H \xi_h^t + e\sum_{t \in I_2}\sum_{h=1}^H \zeta_h^t. \tag{6}$$

Note that we have the following inequalities:

$$\sum_{t \in I_2} \sum_h \frac{1}{N_h^t(s_h^t, a_h^t)} \le \sum_{t \in [T]} \sum_h \frac{1}{N_h^t(s_h^t, a_h^t)} = \sum_{(h,s,a)} \sum_{i=1}^{N_h^T(s,a)} \frac{1}{i} \le \sum_{(h,s,a)} (1 + \ln(N_h^T(s,a)))$$
$$\le HSA(1 + \ln(T/SA)),$$

and

$$\sum_{t \in I_2} \sum_h \frac{1}{\sqrt{N_h^t(s_h^t, a_h^t)}} \le \sum_{t \in [T]} \sum_h \frac{1}{\sqrt{N_h^t(s_h^t, a_h^t)}} = \sum_{(h,s,a)} \sum_{i=1}^{N_h^T(s,a)} \frac{1}{\sqrt{i}}$$
$$\le 2 \sum_{(h,s,a)} \sqrt{N_h^T(s,a)} \le 2\sqrt{HSATH}.$$

Denote $\epsilon := \epsilon_{\text{app}}(\delta/4)$ for simplicity. Under event $E$, we have

$$\sum_{t \in I_2} \sum_h \frac{1}{\widetilde{N}_h^t(s_h^t, a_h^t)} \le \sum_{t \in I_2} \sum_h \frac{1}{N_h^t(s_h^t, a_h^t) - \epsilon} \le \sum_{t \in [T]} \sum_h \frac{1}{N_h^t(s_h^t, a_h^t)}$$
$$\le HSA(1 + \ln(T/SA)),$$

and

$$\sum_{t \in I_2} \sum_h \frac{1}{\sqrt{\widetilde{N}_h^t(s_h^t, a_h^t) - \epsilon}} \le \sum_{t \in I_2} \sum_h \frac{1}{\sqrt{N_h^t(s_h^t, a_h^t) - 2\epsilon}} \le \sum_{t \in [T]} \sum_h \frac{1}{\sqrt{N_h^t(s_h^t, a_h^t)}}$$
$$\le 2H\sqrt{SAT}.$$

Plugging these inequalities and Lemma C.4 into Equation (6), we have that, under event $E$,

$$\sum_{t \in I_2} \Delta_1^t \le 10H^2\sqrt{SAT \ln(8SAHT/\delta)} + 9H^3 S^2 A \ln(THS^2 A/\delta) \ln(eT/SA) + 8\epsilon H^2 S^2 A \ln(eT/SA). \qquad (7)$$

Finally, under event $E$, we also have

$$\sum_{t \in I_1} \Delta_1^t \le 3HSAH\epsilon_{\text{app}}(\delta/4), \qquad (8)$$

due to Lemma C.5.

Using Equation (7) and Equation (8) completes our proof. $\qquad \square$

## D. Lower Bounds

We will first prove a lower bound for $\rho$-TV-stable algorithms for multi-armed bandits (MABs) and construct a reduction from any $\rho$-TV-stable RL algorithm to $\rho$-TV-stable MAB algorithms.

**Lemma D.1.** *Let $\mathbf{A}$ be any $\rho$-TV-stable, sub-linear algorithm for $k$-armed bandits. Fix any non-optimal arm $a$ and let $\Delta_a$ be the gap between the expected mean of arm $a$ and of the optimal arm. Then, for sufficiently large $T$, $\mathbf{A}$ pulls arm $a$ at least $\frac{1}{16\rho\Delta_a}$ many times with probability at least $1/2$, for some absolute constant $c$.*

*Proof of Lemma D.1.* Fix any $\mathbf{A}$ be any $\rho$-TV-stable, sub-linear algorithm for $k$-armed bandits. Fix any $k$-armed bandit instance $P$ where the reward distribution for each arm is a Bernoulli. Let $a$ be any sub-optimal arm of this bandit instance, with the sub-optimality gap (i.e., the difference between the mean reward of the optimal arm and the mean reward of the sub-optimal arm $a$) $\Delta_a$. Let $N_a$ be the number of times out of $T$ rounds that $\mathbf{A}$ pulls arm $a$. Consider the event:

$$E = \{N_a < t_a\} \text{ where } t_a := \frac{1}{16\rho\Delta_a}.$$

We need to show:

$$\Pr_{\mathbf{A},P}(E) < \frac{1}{2}. \tag{9}$$

*Remark* D.2. Our proof strategy broadly follows the strategy used in (Shariff & Sheffet, 2018, Claim 14), with two key distinctions: (i) we use Lemma A.6 in the place of the version of (Karwa & Vadhan, 2017, Lemma 6.1) that (Shariff & Sheffet, 2018) used there, as we deal with TV-stability instead of DP, (ii) we believe our proof is clearer and more rigorous, as we describe our detailed reduction from the claim to be proved to Lemma A.6.

In particular, we will prove **??** via two arguments: first, we show a similar inequality but for a new yet similar MAB instance $Q$, and then connect this inequality to **??** via Lemma A.6. In particular, consider a new $k$-armed bandit instance $Q$ that has the same mean rewards for all $k$ arms as $P$, except only that the mean reward for arm $a$ is by $\Delta_a$ larger than the mean reward for the optimal arm of $P$. The expected regret of $\mathbf{A}$ in $Q$ is at least $\Pr_Q(E)(T - t_a)\Delta_a$, as it follows from that under $E$, the number of times an sub-optimal arm of $Q$ were pulled is at least $T - t_a$, and the sub-optimality gap of a sub-optimal arm in $Q$ is at least $\Delta_a$. For sufficiently large $T$, we have $T - t_a \geq \frac{T}{2}$. At the same time, the expected regret of $\mathbf{A}$ in $Q$ is at most $T^{1-\alpha}$ for some fixed $\alpha \in (0, 1]$ as $\mathbf{A}$ has sublinear regrets. Thus, we have

$$\Pr_{\mathbf{A},Q}(E) \leq \mathcal{O}(\frac{1}{T^\alpha \Delta_a}) \leq \frac{1}{4}, \tag{10}$$

for sufficiently large $T$.

Now consider the following mechanism $M_{\mathbf{A}|\mathcal{D}} : \{0,1\}^{t_a} \to \{0,1\}$ that maps from a $t_a$-bit binary string to a binary output, conditioned on data $\mathcal{D}$ that is independent of $\mathbf{A}$. In addition, the mechanism depends on the learning algorithm $\mathbf{A}$.

Given a sample $\underline{x} \in \{0,1\}^{t_a}$ and a data $\mathcal{D} \in \{0,1\}^{(k-1)(T-t_a)}$, here is how $M_{\mathbf{A}|\mathcal{D}}$ produces the output on $\underline{x}$. Initialize $U = \underline{x}$. We iterate over $T$ steps. In each step: if $\mathbf{A}$ pulls $a$ and $U$ is empty, we terminate and output 1; if $\mathbf{A}$ pulls $a$ and $U$ is not empty, we remove a sample from $U$ and use it as a reward sample for $a$ and feed it back to $\mathbf{A}$ and move to next step and repeat; if $\mathbf{A}$ does not pull $a$, extract from $\mathcal{D}$ a corresponding reward for the pulled arm, move to the next step and repeat. If the iteration loop over $T$ survives all the way down to the last iteration $T$, output zero, i.e., if after $T$ iterations we have not outputted 1, output 0.

We have the following properties:

- $\Pr_{\mathbf{A},P}(E|\mathcal{D}) = \Pr_{\mathbf{A},\underline{X}\sim P_a}(M_{\mathbf{A}|\mathcal{D}}(\underline{x}) = 1)$, by the construction of $M$,

- $M_{\mathbf{A}|\mathcal{D}}(\cdot)$ is $\rho$-TV-stable, because the output of $M_{\mathbf{A}|\mathcal{D}}(\underline{x})$ is simply a deterministic map from the output of $\mathbf{A}$ on $(\mathcal{D}, \underline{x})$ to $\{0,1\}$ and $\mathbf{A}$ is $\rho$-TV-stable.

Thus, we now apply Lemma A.6 to $M_{\mathbf{A}|\mathcal{D}}$, we have that for any $\mathcal{D}$,

$$\begin{aligned}
\Pr_{\mathbf{A},P}(E|\mathcal{D}) - \Pr_{\mathbf{A},Q}(E|\mathcal{D}) &= \Pr_{\mathbf{A},\underline{X}\sim P_a}(M_{\mathbf{A}|\mathcal{D}}(\underline{x}) = 1) - \Pr_{\mathbf{A},\underline{X}\sim Q_a}(M_{\mathbf{A}|\mathcal{D}}(\underline{x}) = 1) \\
&\leq 4t_a \rho \mathrm{TV}(P_a, Q_a) \\
&= 4t_a \rho \Delta_a
\end{aligned}$$

Hence,

$$\begin{aligned}
\Pr_{\mathbf{A},P}(E) - \Pr_{\mathbf{A},Q}(E) &= \sum_{\mathcal{D}} \Pr_{\mathbf{A},P}(E|\mathcal{D})P(\mathcal{D}) - \sum_{\mathcal{D}} \Pr_{\mathbf{A},Q}(E|\mathcal{D})Q(\mathcal{D}) \\
&= \sum_{\mathcal{D}} \Pr_{\mathbf{A},P}(E|\mathcal{D})\Pr(\mathcal{D}) - \sum_{\mathcal{D}} \Pr_{\mathbf{A},Q}(E|\mathcal{D})\Pr(\mathcal{D}) \\
&\leq 4t_a \rho \Delta_a \sum_{\mathcal{D}} \Pr(\mathcal{D}) \\
&= 4t_a \rho \Delta_a = \frac{1}{4}
\end{aligned}$$

where the second inequality follows from that $P$ and $Q$ have the same arm distributions except for arm $a$ and $\mathcal{D}$ does not contain rewards for arm $a$. Now combine the above inequality with Equation (10), we have

$$\Pr_{\mathbf{A},P}(E) \leq \frac{1}{4} + \frac{1}{4} = \frac{1}{2}.$$

$\square$

**Corollary D.3.** *The expected regret of any $\rho$-TV-stable algorithms for $k$-armed bandits is $\Omega\left(\frac{k}{\rho}\right)$.*

*Proof of Corollary D.3.* It follows from Lemma D.1 that, for any $k$-armed bandit where the first arm is the optimal arm and all the other $k-1$ arms $\{2,\ldots,k\}$ are sub-optimal with sub-optimal gaps $\{\Delta_2,\ldots,\Delta_k\}$, we have

$$\mathbb{E}[R_{\mathbf{A}}(T)] = \sum_{i=2}^{k} \Delta_i \mathbb{E}_{\mathbf{A}}[N_i] \geq \sum_{i=2}^{k} \Delta_i t_i \Pr(N_i \geq t_i) \geq \frac{k-1}{32\rho}.$$

$\square$

The next lemma shows that there exists a class of MDP instances with $S$ states, $A$ actions and $H$ horizon where any $\rho$-TV-stable RL algorithms can be simulated by $S$ $\rho$-TV-stable MAB algorithms.

**Lemma D.4.** *There exists a class of MDP instances with $3S$ states, $A$ actions and $H$ horizon where any $\rho$-TV-stable RL algorithms can be simulated by $S$ $\rho$-TV-stable $A$-armed bandit algorithms where the scale of the rewards for each MAB is $H$.*

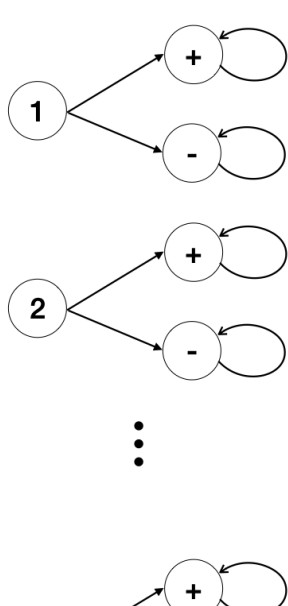

*Figure 1.* Hard MDPs.

*Proof of Lemma D.4.* To prove the hardness result, consider the following class of MDPs in Figure 1. An MDP in this class starts uniformly at random in one of $n$ initial states. From each such state, any action from a set of $A$ actions will lead to only one of the two absorbing states where the agent will stay until the end of an episode. Such an MDP can be viewed as $n$ $A$-armed bandits in parallel.

Let $\mathbf{A}$ be any $\rho$-TV-stable algorithm. Let $\{s_1^t\}_{t \in [T]}$ be $T$ initial states, representing $T$ users $\mathcal{U}$. For any $s \in \mathcal{S}$, let $\mathbf{A}_s$ be all the components of the output of $\mathbf{A}$ that correspond to $s_1^t = s$. Suppose all the episodes $t$ where $s_1^t = s$ are $t_1, \ldots, t_{T_s}$. Fix any event $E_s \subseteq \Pi^{T_s}$. Let $E = \{e \in \Pi^T : (e_{t_1}, \ldots, e_{t_{T_s}}) \in E_s\}$. By marginalization, we have

$$\Pr(\mathbf{A}_s(\mathcal{U}) \in E_s) = \Pr(\mathbf{A}(\mathcal{U}) \in E)$$

Thus, let $\mathcal{U}'$ be a neighbor of $\mathcal{U}$ (they differ by one user), we have

$$\Pr(\mathbf{A}_s(\mathcal{U}) \in E_s) = \Pr(\mathbf{A}(\mathcal{U}) \in E) \leq \Pr(\mathbf{A}(\mathcal{U}') \in E) + \rho = \Pr(\mathbf{A}_s(\mathcal{U}') \in E_s) + \rho,$$

where the inequality is due to that $\mathbf{A}$ is $\rho$-TV-stable. Therefore, the above inequality implies that $\mathbf{A}_s$ is also $\rho$-TV-stable.

In general, $\mathbf{A}_s$ may not be equivalent to an MAB algorithm, as it also uses the information of $\mathbf{A}$ pulling actions for states other than $s$. Thus, $\mathbf{A}_s$ may be more powerful than an MAB thus it can break the lower bound in Corollary D.3. However, in the class of MDPs we construct, the information of $\mathbf{A}$ pulling actions for state $s'$ gives zero information the rewards and next states for state $s \neq s'$. Thus, $\mathbf{A}_s$ is a valid MAB algorithm. $\square$

*Remark* D.5. Note that the MDP class by (Vietri et al., 2020) use the same absorbing states for all initial states, while our MDP class use separate absorbing states for different initial states. We believe this is an analysis gap in (Vietri et al., 2020), though the fix is simple (using our MDP class in Figure 1 instead). The reason is that their MDP class may not give a valid reduction to MAB algorithms. In particular, since all the initial states share the same absorbing states, an RL algorithm can learn this fact, figure out that the absorbing state $+$ is better than the absorbing state $-$, and when it encounters a new initial state $s$, it will bias toward selecting actions that lead to the absorbing state $+$. This will break the lower bound in Corollary D.3.

*Proof of Theorem 4.4.* The first term in our lower bound is the lower bound for any RL algorithm (Auer et al., 2008), thus is applicable to any $\rho$-TV-stable algorithms. We only need to prove that $\frac{HSA}{\rho}$ is also a lower bound. We use the same MDP class as Figure 1 in the proof of Lemma D.4. Any RL algorithm must need to solve $S$ independent $A$-armed bandit problems of scale $H$. By Lemma D.4, any $\rho$-TV-stable RL algorithm can be simulated by $S$ $\rho$-TV-stable MAB algorithms that solve $S$ independent MAB problems. By Corollary D.3, each of them must incur a regret of $\frac{HA}{\rho}$, where $H$ is the reward scale of each MAB. Solving an MDP in our MDP class requires to solving $S$ independent MAB problems, thus the total regret must incur at least $\frac{SHA}{\rho}$. $\square$

