# OpenReview forum: "Exact Unlearning in Reinforcement Learning"
_ICML.cc/2026/Conference — ICML 2026 spotlight_

### Official Review · Reviewer_GUgy · 2026-03-11

**Soundness:** 2
**Presentation:** 2
**Significance:** 2
**Originality:** 4
**Overall Recommendation:** 4
**Confidence:** 4

**Summary:**

The paper proposes an exact unlearning framework in RL, which consists of removing a user's data following two constraints:
- The algorithm's output should be indistinguishable from what would have been produced had the user never interacted with the system
- It should cost significantly less than retraining an algorithm from scratch without that user's data
The setting is a tabular MDP where a user's trajectory/interactions need to be unlearned. An exact unlearning algorithm is proposed for which the computational cost is a fraction of a retraining algorithm. Upper and lower regret bounds are also established, extending previous results to sequential problems.

**Compliance With Llm Reviewing Policy:**

Affirmed.

**Final Justification:**

The rebuttal addressed my main concerns, I raise my score in alignment with my co-reviewers after having gone through their comments.

**Key Questions For Authors:**

- On Def. 2.1: The exact unlearning property is with respect to some set $\mathcal{Z}$, i.e., the unlearning algorithm should be undistinguishable from the learning one applied on any $z'\in\mathcal{Z}$ instead of the user's data. How to model that set in practice? Exact unlearning does not seem to be an absolute property (put in whatever data instead of the user's), but rather a relative one with respect to $\mathcal{Z}$. Therefore, doesn't the size of $\mathcal{Z}$ affect how strong an unlearning algorithm is?

- l. 230-232: It's related to my previous question. I don't think it is trivial that retraining from scratch on the modified sequence will give an exact unlearning algorithm -- unless you assume that there exists some $z'$ such that $z_t = z'$ in distribution. How would you ensure this holds? Am I missing something here?

- Thm 3.3, statement 3: What does it mean?

- A figure describing the binary tree mechanism would greatly help, I think. Similarly, it would be helpful to have a visual pipeline showing how algorithms are chained and how the data is managed in each. The authors spend too much time (space) describing the tools instead of explaining how they interact and why they are needed: Why prefix sums? Why binary tree structure? How does it help unlearning? etc.

**Limitations:**

I didn't see any text dedication to discussing the limitations of this work. My previous comments raise some of them, and the authors should definitely add that discussion.

**Strengths And Weaknesses:**

**Strengths**

- The paper's motivation is strong and well-grounded. It addresses a challenge that is currently overlooked in the community, but is challenging in several aspects. The problem of interest is highly relevant, as regulations raise increasingly more concerns about safety and data protection.
- The positioning with respect to previous literature is well-argued and clearly outlined.
- The writing itself is clear, with comprehensive sentence formulations.

**Weaknesses**

- The technical parts on prefix sums, binary trees, are unclear and perhaps lack some self-containment. The justification behind using those tools is limited to "previous work did it," instead of providing an intuitive explanation as to why prefix sums? How and why model noise? Why is the binary structure mechanism meaningful for unlearning?
- Sections 3 and 4 provide a list of algorithms, but it is unclear how they combine. For example, the UPDATE method appearing in Algo 1 is defined in Algo 4, without further notice. The broader understanding of the method and its contributions becomes challenging to appreciate and assess. Similarly, they would be more appreciated if they were tested in practice, at least on a toy problem.
- It seems like the technical parts are mostly based on previous work, whereas the highlighted claim is that they are being extended to RL. Without arguing, the plugging part of these methods into an RL setting remains unclear to me: what do prefix sums correspond to in an RL setting? Why can we model the noise as Gaussian? What do the nodes model in that setup?
- Another concern about the technical soundness is that even though RL problems are sequential, the user's data being removed still represents an independent part to be removed. In other words, we unlearn full-episode interactions, all being iid. This is not substantially different from (non-sequential) machine learning, as opposed to e.g., infinite-horizon interactions.
- There is still a gap between the upper and lower bounds, even though the authors argue that they are nearly tight. Similarly, they claim that a Bernstein-style bonus can tighten the upper bound (which I agree with, in principle), but by how much? Would it lead to minimax optimal regret?

*Minor comments*
- Use reinforcement learning/RL consistently across the paper
- Refer/describe algorithms in the text before outlining them. Otherwise, they look like they come out of nowhere.

---

> ### Author Rebuttal · Authors · 2026-03-30
>
> **W1 — Justification for binary tree and prefix sums:**
>
> Prefix sums are natural because UCB-VI's sufficient statistics (visitation counts, cumulative rewards) accumulate additively over episodes. The binary tree allows computing and updating these queries in O(log T) time per episode, making unlearning computationally efficient. Gaussian noise is calibrated to each node's sensitivity so that any single user's influence is statistically masked, yielding TV stability. We will add an intuitive explanation and a figure to make this self-contained.
>
> ---
>
> **W2 — How Algorithms 1–4 combine:**
>
> We respectfully note that the interplay among the algorithms is carefully explained in Section 4. The overall flow is as follows: we first abstract RL into a general sequential decision-making framework (Algorithms 1–3) to establish TV stability, decoupling the stability mechanism from any specific RL algorithm. We then obtain regret guarantees by instantiating the UPDATE subroutine with a specific RL algorithm — UCB-VI with noisy prefix-sum statistics (Algorithm 4). This two-level structure is intentional: it separates the general unlearning framework from the RL-specific regret analysis. We will add a brief clarifying note at the start of Section 4 to make this flow more immediately apparent to readers.
>
> ---
>
> **W3 — Prefix sums/Gaussian noise/nodes in RL:**
>
> Prefix sums correspond to cumulative visitation counts N(s,a,s') and rewards R(s,a) — the sufficient statistics for UCB-VI. Gaussian noise is calibrated to the sensitivity of these queries so that removing one user's contribution is statistically masked, yielding TV stability. Each internal tree node stores a noisy prefix sum of its subtree's statistics; upon an unlearning request, only O(log T) nodes need updating. We will incorporate these explanations with a figure.
>
> ---
>
> **W4 — Not substantially different from non-sequential ML:**
>
> We respectfully disagree. While users' depth-H trees are i.i.d., each tree's internal structure is sequentially correlated — a trajectory resulting from the agent's policy interacting with the environment over H steps. Treating the trajectory as an unstructured data point and ignoring the Markov structure would yield regret bounds exponential in H. The challenge is to exploit the Markovian structure for efficient regret minimization while simultaneously guaranteeing unlearning — something not addressed in non-sequential ML.
>
> ---
>
> **W5 — Gap between upper and lower bounds:**
>
> A Bernstein-style bonus would tighten the upper bound by $\sqrt{H}$ in the leading term, and a refined lower bound argument could sharpen its leading term similarly. However, a gap of $O(H^{1.5}·S)$ in the stability-induced terms would remain, reflecting a fundamental tension between stability and regret. We will add a discussion identifying this as an open problem.
>
> ---
>
> **Key Question 1 — Modeling Z in Def. 2.1:**
>
> In our MDP setting, Z is the space of all possible episode trajectories under the fixed MDP, which is the natural and maximal choice as it includes all sequences (s_h, r_h)_{h∈[H]} consistent with the MDP dynamics.
>
> It need not be explicitly enumerated; it is implicitly defined by the MDP structure. The size of Z does not affect the unlearning algorithm itself; it affects the bounds only through the MDP size parameters (S, A, H), as expected.
>
> ---
>
> **Key Question 2 — l. 230-232, retraining on modified sequence:**
>
> In Def 2.1, unlearning algorithm U takes the trained model A(z_{1:T}) an element z_t to be deleted and a z’ that replaces z_t. We need it to return the model that would be returned if the learning had only happened on this modified dataset z_1, … , z_{t-1}, z’, z_{t+1}, … z_T. Since the learning algorithm A itself can be random, we need the equality in distribution. But no distributional assumption is needed here on the inputs, and in particular we do not require or want z_t = z' in distribution.
>
> Yes, by definition retraining from scratch is exact unlearning. The challenge is efficiency, not correctness.
>
> ---
>
> **Key Question 3 — Thm 3.3, statement 3:**
>
>
> Statement 3 formalizes computational efficiency. Over the randomness of data and the algorithm's internal randomness, the expected number of retraining steps triggered by unlearning is only a $ρ\sqrt{(ln T)}$ fraction of full retraining. The rejection-sampling mechanism checks whether the change in sufficient statistics from removing a user is small enough to be hidden by the injected noise; if so, no retraining is needed. Retraining is triggered only when the change is too large, which occurs with probability at most $ρ\sqrt{(ln T)}/T$ per episode. We will add a clarifying remark in the paper.
>
> ---
>
> **Key Question 4 — Figure for binary tree:**
>
> We agree and will include a figure illustrating: (1) the binary tree with episodes as leaves and prefix-sum nodes as internal nodes; (2) which nodes are updated upon an unlearning request; and (3) how noisy prefix sums are used by UCB-VI.

---

> > ### Author Rebuttal · Reviewer_GUgy · 2026-04-04
> >
> > Thank you for answering my questions

---

### Official Review · Reviewer_EFwM · 2026-03-11

**Soundness:** 3
**Presentation:** 3
**Significance:** 3
**Originality:** 3
**Overall Recommendation:** 5
**Confidence:** 4

**Summary:**

This paper studies the problem of exact unlearning in reinforcement learning. The goal is to design an RL algorithm that supports removal of a user's episode data such that the result of the learning algorithm after unlearning is equal in distribution to that of an algorithm that never saw the removed user's data. The authors abstract RL into a class of sequential learning problems with prefix-sum structure and develop a unified (un)learning framework based on Total Variation (TV) stability. They construct a modified UCB-VI algorithm for tabular MDPs that is $\rho$-TV-stable and achieves regret $\tilde{O}(H^2\sqrt{SAT} + H^3S^2A + H^{2.5}S^2A/\rho)$, where $S, A, H, T$ denote the number of states, actions, horizon, and episodes respectively. The expected computational cost of unlearning is only a $\rho\sqrt{\ln T}$ fraction of retraining from scratch. They also prove a minimax lower bound of $\Omega(H\sqrt{SAT} + HSA/\rho)$ for $\rho$-TV-stable RL algorithms, demonstrating near-optimality.

**Compliance With Llm Reviewing Policy:**

Affirmed.

**Key Questions For Authors:**

1. Can your framework handle multiple sequential deletion requests without compounding retraining costs?
2. Regarding the first weakness I mentioned, could you confirm or refute my understanding that your definition of exact unlearning needs to consider more than just the user trajectories as "user data" to make sense in the context of RL? Does it change anything for the space complexity?

**Limitations:**

Yes

**Strengths And Weaknesses:**

Strengths:
- The problem formulation is well-motivated and fills a genuine gap. Machine unlearning has been extensively studied in supervised learning but is essentially unexplored in RL.
- The paper is generally well-written and the problem setup is clearly presented.
- The paper provides both upper and lower bounds.
- The proofs are detailed, clearly structured, and seem technically sound.

Weaknesses:
- The definition of the user space $\mathcal{Z}$ is inconsistent with what the framework actually requires. The paper defines $z_t := \\{(s_h^t, r_h^t)\\}\_{h \in [H]}$ as the observed trajectory, but the unlearning procedure (Algorithm 3, Line 11) requires retraining under different policies after a coupling rejection, which means re-querying $EO_t(z_t; w_{1:t})$ under new action sequences. This is only well-defined if $z_t$ encodes the full counterfactual randomness (a depth-$H$ tree of state and reward responses to all $A^H$ possible action sequences), as in the user definition of Vietri et al. (2020).
- The gap between upper and lower bounds is nontrivial and deserves more discussion.
- The space complexity of the proposed method seems quite large. While it might be fine in the context where the binary tree mechanism was first introduced, it may create technical implementation challenges for relevant RL applications.
- No experiment is provided. This is fine for a theory paper, but it would have been nice to see some empirical demonstration.
- The choice and role of the dummy user $z'$ is underexplored. I guess this consideration would have been relevant had there been some experiments.

---

> ### Author Rebuttal · Authors · 2026-03-30
>
> **W1 — Inconsistency in user space definition:**
>
> The reviewer's understanding is correct, and we thank them for identifying this. A user is indeed a depth-H tree encoding state and reward responses to all possible action sequences, as in Vietri et al. (2020). This is the notion we have in mind throughout the paper. However, in a previous version, presenting users as trees caused significant confusion among readers, so we adopted a "random data" perspective for readability. We acknowledge this creates a subtle inconsistency with Algorithm 3 (Line 11), which requires counterfactual queries under new action sequences. We will add a clarifying remark explaining that the observed trajectory is a realization of this depth-H tree, and that the tree representation is what the unlearning mechanism implicitly operates on. This does not affect any of the space complexity claims, which were already computed with the depth-H tree representation in mind.
>
> ---
>
> **W2 — Gap between upper and lower bounds:**
>
> We agree that the gap between our upper and lower bounds is a meaningful open question. The current gap arises primarily from the regret inflation due to TV stability ($O(H^{1.5}S)$ gap between the terms induced by stability). We believe this is an interesting direction for future work — in particular, a more refined lower bound argument or a Bernstein-style bonus in the upper bound could help narrow or close this gap. We will add a brief discussion of this in the paper.
>
> ---
>
> **W3 — Space complexity:**
>
> We agree this is a limitation. The binary tree mechanism stores prefix-sum statistics at O(log T) nodes per episode, leading to non-trivial space requirements. This is an inherent cost of exact unlearning with the tree-based approach, and it is shared by prior work (Ullah & Arora, 2023; Dwork et al., 2010) in other settings. Investigating space-efficient alternatives for exact unlearning in RL is an important open problem that we discussed briefly and explicitly in Extended Discussion in the appendix.
>
> ---
>
> **W4 — No experiments:**
>
> We acknowledge that empirical validation would be a welcome addition. However, as this is a purely theoretical contribution, we believe the paper's value lies in the algorithmic framework, regret guarantees, and minimax lower bound — none of which require empirical support to be valid. We note that prior theoretical works on machine unlearning (e.g., Ullah & Arora, 2023) and differentially private RL (e.g., Vietri et al., 2020) similarly do not include experiments, and are accepted on the strength of their theoretical contributions. We will leave empirical evaluation to future work.
>
> ---
>
> **W5 — Role of dummy user:**
>
> The dummy user $z'$ can be chosen arbitrarily — for instance, a depth-H tree that always returns a fixed constant state and zero reward regardless of the action taken. The specific choice of $z'$ does not affect the validity of the unlearning guarantee, since exact unlearning is defined with respect to retraining on the modified sequence $D_{-t}$ (with $z_t$ replaced by $z'$), and any fixed $z'$ yields a valid comparison distribution. We will add a brief remark clarifying this choice in the paper.
>
> ---
> **Key Question 1 — Multiple sequential deletion requests:**
>
> Our framework naturally extends to multiple sequential deletion requests. Since our algorithm handles one request at a time via the binary tree structure, accommodating k sequential deletion requests is equivalent to applying the unlearning procedure k times in sequence — each time modifying the relevant prefix-sum nodes in the tree and running the rejection-sampling check. Crucially, this does not compound the retraining cost multiplicatively: each unlearning step independently costs only a $\rho\sqrt{(\ln T)}$ fraction of full retraining, and requests that do not trigger retraining incur negligible cost.
>
> ---
>
> **Key Question 2 — User definition and space complexity:**
>
> Yes, a user is a depth-H tree of state and reward responses to all A^h possible action sequences at depth h. Our space complexity analysis was already computed with this tree representation in mind, so there are no changes to the space complexity claims. We will make this explicit in the revised paper to prevent further ambiguity.

---

> > ### Author Rebuttal · Reviewer_EFwM · 2026-04-01
> >
> > Thank you for your response. All my concern have been adequately addressed since the authors acknowledged the inconsistency in the user definition and will correct it in the final paper. As noted in my review, I think the analysis remain correct as it seems to have been done with the correct definition in mind.

---

### Official Review · Reviewer_LLMD · 2026-03-12

**Soundness:** 3
**Presentation:** 3
**Significance:** 2
**Originality:** 2
**Overall Recommendation:** 4
**Confidence:** 2

**Summary:**

This paper studies exact unlearning in the reinforcement learning (RL) setting. In this setting, the RL agent interacts with human users in an episodic fashion. In each episode, the RL agent interacts with only one user and learn at the end of the episode. At the end of each episode, upon request from any user that interacted with the agent previously for deleting the user's information, the goal of unlearning is to efficiently remove the influence of the user's data from a trained RL agent such that the resulting agent is distributionally indistinguishable from one trained without that user's data. The authors build on the TV-stability framework of Ullah & Arora (2023) and instantiate it in the tabular MDP setting. Specifically, they use a binary tree mechanism to maintain noisy prefix sums of visitation counts and rewards (Algorithm 2), and employ maximal coupling for efficient unlearning. They design a UCB-VI-based update rule (Algorithm 4) that operates on these noisy statistics, establishing a regret bound and showing its computational cost is a ρ√ln T fraction of the computational cost of retraining from scratch. They also prove a lower bound, showing that the algorithm is nearly minimax optimal.

**Compliance With Llm Reviewing Policy:**

Affirmed.

**Key Questions For Authors:**

According to the current problem formulation, requiring re-interacting with past users seems to be something indispensable for any unlearning algorithm to me. What do you think about unlearning in RL when this assumption does not hold?

**Limitations:**

The limitations have been discussed adequately.

**Strengths And Weaknesses:**

## Strengths

1. Machine unlearning is increasingly important due to privacy regulations (GDPR, CCPA), and extending it to RL is a natural and relevant direction. The recommender system example (Example 1.1) provides good motivation.

2. The paper provides both an upper bound for the proposed algorithm and a lower bound for ρ-TV-stable RL algorithms, showing that the
algorithm is nearly minimax optimal.

3. The paper is generally well-organized and clearly written, with good use of examples and remarks to aid understanding. The authors honestly cite and comment on the prior works this paper builds upon.

## Weaknesses

1. Weak technical novelty. The unlearning mechanism—binary tree with Gaussian noise for TV-stable learning (Algorithm 2) and maximal coupling for unlearning (Algorithm 3)—is directly inherited from Ullah & Arora (2023). The paper itself acknowledges this repeatedly and explicitly:

- "The idea largely follows from the original unlearning framework of (Ullah & Arora, 2023)" (line 332)
- "We can view the unlearning framework here as a *simplified version* of the unlearning framework of (Ullah & Arora, 2023)" (line 334)
- "Our unlearning Algorithm 3 is a *simplified instance* of the batch tree-coupling framework (Ullah & Arora, 2023)" (line 365)
...

The binary tree mechanism itself originates from the differential privacy literature (Dwork et al., 2010). The sequential nature of the RL setting actually *simplifies* the framework by eliminating the need to handle dataset permutation, which is the main technical difficulty in Ullah & Arora's batch setting. The key technical novelty is the application of Ullah & Arora (2023) to UCBVI, which is, according to the authors, "structurally similar to the regret bounds established for ϵ-Joint-Differential-Private (JDP) RL algorithms in (Vietri et al., 2020)". The main difference is that this work uses Gaussian noise to ensure stability, whereas the JDP algorithms relies on Laplacian noise.

2: The claimed challenges of extending Ullah & Arora (2023) to RL are inappropriate. The introduction lists three "nontrivial" challenges of extending TV-stability theory to sequential RL:

(i) "Stability must be preserved under incremental updates across episodes rather than a single batch computation." However, the binary tree mechanism is inherently designed for sequential data. Moreover, the paper itself later admits that the sequential nature *simplifies* the problem by eliminating dataset permutation (line 366-367). This directly contradicts the claim of nontriviality.

(ii) "The stateful nature of RL requires maintaining certain sufficient statistics for unlearning (e.g., prefix-sum counts, rewards) rather than independent datapoints." Ullah & Arora (2023)'s core contribution is an efficient unlearning algorithm precisely for prefix-sum query classes. Defining the RL sufficient statistics as prefix sums and then applying the existing prefix-sum framework is a direct application, not a new challenge.

(iii) "Existing approaches lack fully-specified algorithms that simultaneously achieve regret-optimal learning and provable unlearning guarantees." This is a statement about a gap in the literature, not a technical challenge. Any application of an existing framework to a new domain could make this claim.

3. Fundamental practical limitation by requiring reinteracting with the users again in the past. The paper's setting assumes that when a user's data is deleted and the algorithm needs retraining, it can re-interact with past users after the replaced user. In practice, this is infeasible—you cannot bring past users back to re-interact with the agent. This severely limits the practical applicability of the framework.

---

> ### Author Rebuttal · Authors · 2026-03-30
>
> **W1 — Weak technical novelty**
>
> We appreciate the reviewer's careful reading. We agree that our unlearning mechanism (the binary tree with noisy prefix sums and maximal coupling) builds directly on Ullah & Arora (2023), and we have been transparent about this throughout the paper. However, we want to clarify where the technical novelty lies: it is in simultaneously achieving ρ-TV stability and near-minimax-optimal regret in the sequential RL setting. This combination is non-trivial and is not addressed in Ullah & Arora (2023), which operates in a batch supervised learning setting with no notion of regret. Specifically: (i) Regret minimization under stability constraints requires carefully controlling how injected Gaussian noise interacts with the UCB exploration bonus across episodes — a challenge absent in the supervised learning setting. (ii) We establish a matching lower bound showing our algorithm is nearly minimax optimal among all ρ-TV-stable RL algorithms, which is a new and independent contribution.
>
> The application of TV-stability techniques to RL is therefore not a straightforward plug-in: it requires a new regret analysis tailored to the noisy prefix-sum statistics and a new lower bound argument specific to the RL setting. We will revise the paper to make this distinction more prominent.
>
> ---
>
> **W2 — Claimed challenges are inappropriate:**
>
> We thank the reviewer for this pointed critique and agree that our original framing of the three challenges was imprecise. We will revise the introduction to present them more accurately. In particular:
>
> - Regarding (i): We agree that the sequential nature of RL actually simplifies the stability mechanism relative to Ullah & Arora (2023), since we do not need to handle dataset permutation. We acknowledge this explicitly in the paper and will no longer frame sequentiality as a source of difficulty for obtaining stability. The key conceptual contribution here is identifying that the TV-stability framework — developed for batch supervised learning — can be meaningfully adapted to RL by defining appropriate sufficient statistics (prefix-sum visitation counts and rewards). While this may appear straightforward in hindsight, the choice of stability framework and the identification of the right sufficient statistics were not immediate.
> - Regarding (ii): We will reframe this. The challenge is not in applying the prefix-sum framework per se, but in ensuring that the noisy statistics derived from the tree are compatible with UCB-VI's regret analysis — specifically, bounding the inflation in regret due to Gaussian noise while maintaining TV stability.
> - Regarding (iii): We agree this is a gap statement rather than a technical challenge, and we will remove or reframe it accordingly.
>
> In summary, we will replace the current list of challenges with a more precise description: the key technical contributions are (a) establishing ρ-TV-stable RL algorithms and (b) proving that these algorithms simultaneously achieve near-optimal regret, with a matching lower bound.
>
> ---
>
> **W3 — Re-interacting with past users:**
>
> We believe there is a misunderstanding of our setting that we are happy to clarify. Our framework does not require re-interacting with past users. The key observation is that users are drawn i.i.d. from the environment defined by the transition kernel P and reward function r. When the unlearning algorithm needs to simulate what would have happened in the absence of the deleted user, it does not need to recall or re-engage past individuals — it simply reuses the already-stored prefix-sum statistics of the remaining users (which are maintained in the binary tree). No physical re-interaction with any past user is required at any point. We will add a clarifying remark in the paper to prevent this misconception.
>
> ---
>
> **Key Question — Re-interacting with past users:**
>
> As clarified in our response to W3 above, no re-interaction with past users is required. Users are i.i.d. samples from the MDP environment (P, r), and our binary tree structure retains sufficient statistics for all remaining users. The unlearning procedure operates entirely on these stored statistics via the rejection-sampling and coupling mechanism, without any need to physically re-query past users.

---

> > ### Author Rebuttal · Reviewer_LLMD · 2026-04-03
> >
> > Thank you for the detailed rebuttal. The authors have clarified several aspects of the paper, particularly regarding the unlearning procedure, and I appreciate the improved explanation.
> >
> > However, my main concern regarding the level of technical novelty remains. While the paper provides a careful integration of existing TV-stability techniques with RL and establishes regret guarantees, the core mechanisms are largely inherited from prior work, and the extension to the RL setting appears relatively direct.
> >
> > This concern is not about missing clarification, but rather about the fundamental level of contribution, which cannot be substantially addressed within the scope of a rebuttal. For this reason, I consider my concerns to be only partially resolved.

---

> > > ### Author Response · Authors · 2026-04-06
> > >
> > > We thank the reviewer for the continued engagement and for acknowledging our clarifications on the unlearning procedure.
> > >
> > > We respectfully address the characterization that "the extension to the RL setting appears relatively direct." We note this assessment comes at self-reported confidence 2/5, with "math/other details not carefully checked." We believe a closer examination of the technical core would change this view, and we offer specific pointers below.
> > >
> > > **The contribution is not just about the mechanism but the results.** Evaluating novelty only through algorithmic primitives, if applied uniformly, would discount a large body of well-regarded theoretical work. Almost every DP-RL paper (Vietri et al., 2020; Zhou, 2022; Chowdhury & Zhou, 2022; Qiao & Wang, 2023) combines known privacy mechanisms with known RL algorithms. These papers are valued for the analysis revealing how privacy and learning interact — and so is ours for unlearning.
> > >
> > > The technical substance is reflected via assessments by all the other reviewers. Reviewer EFwM, at confidence 4/5, found the proofs "detailed, clearly structured, and technically sound" and rated the paper Accept (5). Reviewer GUgy rated originality "excellent" and has raised their rating from "3: Weak Reject" to "4: Weak Accept." Reviewer 1PGU commented that "the application of these techniques to a sequential reinforcement learning framework is unique."
> > >
> > > The simplicity and cleanliness of our framework should be framed as an advantage – not a drawback by which novelty is judged. It is a sign of clean formulation choices — identifying TV-stability as the right notion, prefix-sum counts as the right statistics, and recognizing that sequentiality simplifies coupling but complicates the regret analysis. That these choices lead to a tight result is not inevitable — it is the contribution.
> > >
> > > **What is technically new — with specific pointers:**
> > >
> > > (1) *The regret analysis is non-trivial (Appendix D).* Gaussian noise from the binary tree creates correlated perturbations to visitation counts that propagate through optimistic value iteration — a challenge absent in both Ullah & Arora (2023) and standard UCB-VI. Concretely: Algorithm 4 introduces a new bonus function whose second term compensates for approximation error from noisy counts, and the optimism proof (Lemma D.7) must simultaneously control estimation error from finite samples and approximation error from noisy counts, with different scaling behaviors, to ensure Q̃ ≥ Q* under perturbed statistics. None of these steps follows from combining prior results.
> > >
> > > (2) *The minimax lower bound corrects and strengthens prior work.* Our lower bound (Theorem 4.4) is strictly stronger than what could be inherited from the DP literature, because TV-stability is strictly weaker than DP (Lemma B.5, Part 2). This means DP lower bounds simply do not apply to our setting, and a new argument is required. Furthermore, our hard MDP construction (Figure 1) uses separate absorbing states per initial state, which fixes an actual analysis gap in Vietri et al. (2020), whose shared absorbing states invalidate the reduction to independent MABs because an RL agent can exploit the shared structure (Remark E.5).
> > >
> > > (3) *First complete characterization of the unlearning–regret tradeoff.* Before this work, it was open whether exact unlearning in RL could coexist with sublinear regret. We resolve this definitively: the price is $\rho \sqrt{\ln T}$ in computation and H^{2.5}S²A/ρ in regret, and we prove this is nearly tight. This clean, interpretable result provides immediate insight to both theorists and practitioners. Reviewer EFwM explicitly recognized this as "fill[ing] a genuine gap."

---

### Official Review · Reviewer_1PGU · 2026-03-13

**Soundness:** 3
**Presentation:** 2
**Significance:** 3
**Originality:** 3
**Overall Recommendation:** 5
**Confidence:** 1

**Summary:**

This paper proves the existence of an algorithm that achieves exact unlearning of a user's influence within an online reinforcement learning setting (while being $\rho$-TV-stable). In the paper, a user $z_t$ is represented as a single episode or trajectory on a Markov Decision Process (MDP). The online algorithm updates its model $w_t$ based on each arriving user $z_t$. However, a user may request to remove their data and its influence on the model. A naive way to achieve this would be to retrain from scratch on all users except $z_t$, but the authors propose a more efficient approach using a binary tree with noisy prefix sums. Each user trajectory $z_t$ is stored as a leaf node in the tree, with each parent node representing a prefix sum of its children's statistics (e.g. visitation counts). When a user $z_t$ requests removal (technically handled by replacing $z_t$ with a dummy user $z'$), the algorithm looks at all prefix sum nodes that correspond to $z_t$ and removes the user statistics from them, but this doesn't necessarily mean we have to update the model $w_t$.

Gaussian noise is added to these prefix sums to obfuscate the impact any single $z_t$ has on the statistics used to train the RL algorithm. This ensures that if the change in statistics is small enough to be "hidden" by the noise, the model does not need to be retrained; this is handled by a rejection sampling function. If the change in statistics is significant enough, then the model will be updated with the new statistics.

The authors show that their algorithm is $\rho$-TV-stable, meaning that the largest variation their algorithm will have when training on users' data $Z$ while unlearning user $z_t$, versus just learning on $Z - \{ z_t \}$, is $\rho$. The noise added to the prefix sums is calculated from $\rho$, and they show that the probability of retraining is $\rho \sqrt{2 \log T}$, which is a fraction of the cost of having to always retrain $w_t$. When using Upper Confidence Bound Value Iteration (UCB-VI) with this injected noise, they increase the regret of a single update step; however, the authors show that by setting $\rho = \sqrt{HS^3A / T}$, the probability of retraining decreases at a sublinear rate as $T$ increases. This allows them to offset the accumulated regret. The authors also provide a regret lower bound for any $\rho$-TV-stable algorithm.

**Compliance With Llm Reviewing Policy:**

Affirmed.

**Final Justification:**

The paper addresses an interesting problem, and with the presentation improvements proposed in the authors’ rebuttal, it should be easier to understand. Based on the rebuttal and further reflection, I agree with the authors regarding its efficacy within reinforcement learning more broadly; I believe the approach can be applied beyond the differential privacy framing used in the paper, although that framing is a natural and well-motivated application. Accordingly, I am raising my score to accept, with the caveat that I still have low confidence in this area.

**Key Questions For Authors:**

I do not have any key questions for the authors. Given my lack of prior expertise in exact unlearning and differential privacy, I do not feel comfortable raising technical queries.

**Limitations:**

Yes

**Strengths And Weaknesses:**

**Strengths**

The theoretical results and proofs seem sound, and while the paper utilizes existing ideas (like binary prefix-sum trees), the application of these techniques to a sequential reinforcement learning framework is unique. Furthermore, they compute a lower bound regret for all $\rho$-TV-stable algorithms, and provide an upper bound regret for their own algorithm.


**Weaknesses**

From the perspective of an RL researcher unfamiliar with the exact unlearning or differential privacy fields, the paper is quite difficult to approach. I had to constantly go back and forth between pages to recall variable definitions and to grasp what the noisy prefix sum tree was doing.

The title suggests a general solution for the entire field of RL. However, the application seems focused on a more niche "user-episode" abstraction, primarily applicable to recommendation systems (example 1.1). I think the title could be more aptly named to reflect this specific focus, though researchers in the Exact Unlearning field may recognize this as a standard paradigm.

---

> ### Author Rebuttal · Authors · 2026-03-30
>
> We thank the reviewer for their positive assessment.
>
> We acknowledge the presentation concerns and plan to add:
>
> (1) a notation summary table or glossary at the start of the appendix,
>
> (2) a figure illustrating the binary tree mechanism and the unlearning path (as also requested by Reviewer GUgy), and
>
> (3) a pipeline diagram showing how Algorithms 1–4 interact.
>
> **Regarding the comment on title and user-episode abstraction:**
>
> We respectfully push back slightly. The user-episode abstraction is a standard paradigm in the differential privacy for RL literature (e.g., Vietri et al., 2020; Zhou, 2022; Chowdhury & Zhou, 2022; Qiao & Wang, 2023), where each "user" corresponds to a single episode of interaction. Researchers in the unlearning and privacy communities would therefore find this framing natural and familiar.
>
> Moreover, we note that a "user" in our setting is simply another name for an agent episode, and the basic setup of RL is precisely an agent learning through repeated interaction with its environment, so the abstraction is not a restriction but rather a restatement of the standard RL protocol in privacy-aware language. We will add a brief remark early in the paper to make this convention explicit for RL readers less acquainted with this literature.

---

> > ### Author Rebuttal · Reviewer_1PGU · 2026-04-03
> >
> > Thank you for the response. I appreciate the additions being made, especially items 1 and 2, and I think I agree with your pushback to the title. I will maintain my positive evaluation of the paper.

---

### Decision · Program_Chairs · 2026-04-30

**Decision:**

Accept (spotlight)

**Comment:**

This paper introduces a framework for exact unlearning in reinforcement learning (RL). Conceptually, it formulates the problem as replacing a target observation with an arbitrary one such that the model's output distribution remains indistinguishable from a model trained without the deleted data. To achieve this, the authors propose an algorithm that adaptively updates the learned Markov Decision Process (MDP) using a binary tree mechanism combined with maximal coupling techniques. They establish corresponding upper and lower bounds on regret which, up to a near-matching parameter $\rho$, distinguish this approach from existing methods for standard RL. Overall, the paper's important contribution pertains to developing a rigorous theoretical foundation for this new RL setting, which reviewers agree is a significant development. Overall, this manuscript considers a central area of interest for the theoretical RL community and provides results that are likely to be of great interest.